



# Local scale evaluation of the simulated interactions between energy, water and vegetation in land surface models.

Jan De Pue[1], José Miguel Barrios[1], Liyang Liu[2], Philippe Ciais[2], Alirio Arboleda[1], Rafiq Hamdi[1], Manuela Balzarolo[3], Fabienne Maignan[2], and Françoise Gellens-Meulenberghs[1]

[1]Department of Meteorological and Climatological Research, Royal Meteorological Institute, Belgium
[2]Laboratoire des Sciences du Climat et de l'Environnement, LSCE/IPSL, CEA-CNRS-UVSQ, Université Paris-Saclay, Gif-sur-Yvette, France
[3]Department of Biology, University of Antwerp, Belgium

**Correspondence:** Jan De Pue (jan.depue@meteo.be)

**Abstract.** The processes involved in the exchange of water, energy and carbon in terrestrial ecosystems are strongly intertwined. To accurately represent the terrestrial biosphere in land surface models (LSM), the intrinsic coupling between these processes is required. Soil moisture and leaf area index are two key variables at the nexus of water, energy and vegetation. Here, we evaluated three LSM (ISBA, ORCHIDEE and a diagnostic model, based on the LSA SAF algorithms) in their ability to

simulate the latent heat flux (LE) and gross primary production (GPP) coherently, and their interactions through leaf area index (LAI) and soil moisture. The models were validated using in situ eddy covariance observations, soil moisture measurements and remote sensed LAI. It was found that the diagnostic model performed consistently well, regardless land cover, whereas important shortcomings of the prognostic models were revealed for in herbaceous/dry sites. Despite their different architecture and parametrization, ISBA and ORCHIDEE shared some key weaknesses. In both models, LE and GPP were found to be

oversensitive to drought stress. Though the simulated soil water dynamics could be improved, this was not the main cause of errors in the surface fluxes. Instead, these errors were strongly correlated to errors in LAI. The simulated phenological cycle in ISBA and ORCHIDEE was delayed compared to observations, and failed to capture the observed seasonal variability. The feedback mechanism between GPP and LAI (i.e. the biomass allocation scheme) was identified as a key element to improve the intricate coupling between energy, water and vegetation in LSM.

## 1   Introduction

Terrestrial ecosystems modulate the surface fluxes of heat, water and carbon, and are thereby an essential driver of weather and climate (Pielke et al., 1998). They are a substantial dynamic component of the global carbon budget, with 15% of the global atmospheric $CO_2$ being yearly exchanged through the stomata of leaves and assimilated through photosynthesis (Ciais et al., 2013). Furthermore, the pivotal role of vegetation on the global climate is mediated by its impact on the hydrological cycle

(Falkenmark et al., 2004). Despite its importance in the frame of the global changing climate, large uncertainties remain in the coupling of the energy, water and carbon cycle in the terrestrial biosphere (Piao et al., 2013; Kauwe et al., 2017; Shukla et al., 2019).





Land surface models (LSM) are key tools to quantify these surface fluxes, and to better our understanding of their interactions. They allow the coupled simulation of the fluxes of water, energy and carbon between the surface and the atmosphere, and are

a crucial component of numerical weather models and earth system models. Over the past decades, they have evolved from their initial simple biophysical configuration to include more complex feedback mechanisms, such as soil moisture dynamics, dynamic vegetation, plant phenology etc. (Delire et al., 2020; Fisher and Koven, 2020).

The processes involved in the surface fluxes from the terrestrial biosphere, such as photosynthesis, transpiration, soil hydrology and leaf phenology, are deeply intertwined with each other. Soil moisture and leaf area index (LAI) are two key prognostic

variables at the nexus between energy, water and vegetation processes.

Root zone soil moisture affects the leaf exchange of water and carbon by modulating the stomatal closure (Raschke, 1979). Although the physiological processes involved are well-described, there is a substantial disagreement in the stomatal behaviour across various models (Kauwe et al., 2017). An evaluation of the impact of soil moisture in the CMIP5 models (Taylor et al., 2012) indicated that the LSM were generally oversensitive to drought stress and wet events (Huang et al., 2016). Whereas

several other studies have reported similar outcomes (Piao et al., 2013; Kolus et al., 2019), Rebel et al. (2012) found an underestimation of ORCHIDEE response to drought. Some of the key challenges lie in: the upscaling of leaf-level processes to canopy-scale and ecosystem-scale simulations (Kauwe et al., 2017), the broad range of processes contributing to evapotranspiration (ET), along with numerous feedback mechanisms (Bonan et al., 2014; Fisher and Koven, 2020), and the difficulty to simulate soil moisture dynamics and infiltration itself (Li et al., 2018; Vereecken et al., 2019). Furthermore, the validation of

these simulations is hampered due to the scale mismatch between flux footprint and model grid and the challenge in accurately observing the partitioning of the surface fluxes (transpiration, soil evaporation, canopy intercept evaporation, etc.; Nelson et al., 2020).

Leaf area index is another key variable in terrestrial ecosystem models, as many leaf-scale processes are scaled to canopy-scale surface fluxes, proportional to LAI. Over the past decades, simulations with prognostic LAI have become an established

approach to account for interseasonal variability of the terrestrial vegetation in land surface models (Calvet et al., 1998; Dickinson et al., 1998; Krinner et al., 2005; Gibelin et al., 2006). The coupling of the carbon assimilation to a biomass allocation scheme allows to simulate the variable phenological cycle, and the vegetation response to meteorological forcings. The degree of complexity of this scheme is very variable amongst models, and range from fairly simplistic (e.g. in ISBA or CTESSEL) to advanced, with dedicated phenology modules or non-structural carbohydrate dynamics (e.g. in ORCHIDEE or CLASS).

Previous studies have concluded that LSM are capable of representing the amplitude of the seasonal LAI cycle with reasonable accuracy (Gibelin et al., 2008), but substantial shortcomings are found in the timing of the phenological cycle and the interseasonal variability (Lafont et al., 2012). The disagreement amongst models (and observations) can be attributed to our limited knowledge on biomass allocation, reserve dynamics and below-ground processes (Le Roux et al., 2001; Fatichi et al., 2019). As a consequence of the coupling of the vegetation dynamics, the uncertainty associated with the seasonal cycle of LAI

propagates back to the surface fluxes.

The resulting feedbacks from the coupling of water, energy and vegetation are summarized in Fig. 1. Soil moisture and LAI are state variables which determine the exchange of heat, water and carbon. Through the feedback to soil moisture in prognostic





models, uncertainties in the exchange of heat and water (e.g. latent heat flux (LE), or evapotranspiration) propagate to the carbon assimilation. Inversely, uncertainties in gross primary production (GPP) or the vegetation growth affect the heat and

water fluxes over LAI. Finally, through phenology equations in some models (e.g. ORCHIDEE for grass), soil moisture can also affect LAI directly.

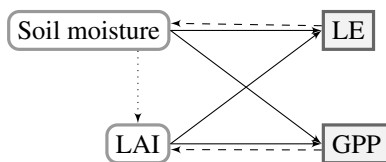

**Figure 1.** Relation (plain lines) and feedbacks (dashed lines) of the state variables and surface fluxes in prognostic LSM. The feedback mechanisms are not present in diagnostic models, and the LAI-Soil moisture relation (dotted line) occurs only in prognostic LSM with dedicated phenology schemes.

To reduce the uncertainties associated with soil moisture and LAI, data assimilation methods have been developed (MacBean et al., 2015; Albergel et al., 2018). Alternatively, diagnostic LSM have been found to be capable of producing robust estimates

of surface fluxes. The surface fluxes in diagnostic LSM are derived from observed state variables, such as remote sensed soil moisture and LAI. They have the advantage that less a priori knowledge or assumptions are required, as less physiological processes need to be accounted for. On the other hand, their application remains restricted to hindcasting of the fluxes, whereas prognostic models are capable of forecasts as well. Though diagnostic models have a less complex model structure, the surface fluxes of energy, water and carbon can be simulated coherently. For example, the GPP predictions in the LSA SAF portfolio

rely on estimates of ET from the same framework (amongst other variables; Martínez et al., 2020). Diagnostic models are frequently used as a reference or validation dataset in LSM studies (e.g. Decharme et al., 2019; Guimberteau et al., 2018). Fewer studies have compared the performance of diagnostic and prognostic models (Yilmaz et al., 2014). Here, we wish to evaluate how coherent the simulated surface fluxes are in comparison to those from prognostic models.

The ever-increasing availability of long-term in situ measurements of energy, water and carbon fluxes from eddy covariance

(EC) tower networks has mediated the evaluation of LSM at local scale (Balzarolo et al., 2014; Napoly et al., 2017; Dirmeyer et al., 2018). In situ EC observations of fluxes and meteorological conditions have been an essential resource in the study of terrestrial ecosystems and the development of LSM. In combination with remote sensed observations of LAI, they provide key insights in the interactions between the surface fluxes and the biosphere.

The objective of this paper is to evaluate the performance of three LSM at local scale. Our focus is the relation between the

surface fluxes (LE, GPP), and important state variables (soil moisture, LAI). This is done by 1) validation and intercomparison of the simulated surface fluxes and prognostic variables in these models, 2) evaluation of the sensitivity of the models to soil moisture and LAI, 3) evaluation of the simulated phenology and its impact on the surface fluxes and 4) comparison of the flux partitioning and water use efficiency of the models. Given the degree of coupling in the current LSM, we try to disentangle the relation between key facets of the terrestrial vegetation in a holistic way.

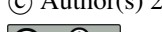



## 2 Materials & Methods

### 2.1 Models

Three well-established models were used to simulate the intrinsically coupled fluxes of water, energy and carbon from terrestrial vegetation: a diagnostic model based on the LSA SAF algorithms (hereafter referred to as DiagMod), ISBA and ORCHIDEE. Each model has a different approach to represent plant phenology. Whereas ISBA has a fairly simple biomass allocation scheme to represent the phenological cycle, ORCHIDEE relies on a dedicated phenology module, and DiagMod is driven by remote-sensed forcing variables, such as LAI.

Simulations were performed for a wide range of hydro-climatic biomes and plant functional types at local scale (i.e. a single grid point). The simulated fluxes were validated using eddy-covariance measurements, and the simulated phenology was compared to remote-sensed observations of LAI.

For adequate intercomparison, the models were configured to run with identical land cover and atmospheric forcing. The vegetation type at each site was derived from ECOCLIMAP 2 (Faroux et al., 2013) and corrected manually to represent the tower footprint area. The sources of the forcing variables are listed in Table 1. ERA5 was used to replace tower variables with large gaps in the time series (e.g. relative humidity; Hersbach et al., 2020). It was verified that the impact of the use of ERA5 instead of local forcings was limited (not shown here). The forcing from ERA5 was linearly interpolated to match the 30 minute temporal resolution from the tower observations.

### 2.1.1 Diagnostic Model (DiagMod)

The diagnostic model used in this study is based on the algorithms used in the LSA SAF products. The LSA SAF algorithm to simulate LE was developed in the framework of EUMETSAT deployment of 'Satellite Applications Facilities' (SAF[1]), and is used in operational products, such as LSA SAF ET. It is a Soil Vegetation Atmosphere Transfer (SVAT) model, largely driven by remote sensed observations of downdwelling long- and shortwave radiation, LAI and albedo. It relies on the Jarvis (1976) approach to calculate the stomatal response to environmental factors.

In operational modus, the observations of the Spinning Enhanced Visible and Infrared Imager (SEVIRI) onboard the Meteosat Second Generation Satellite (MSG) are the primary source of the forcing variables. A more in-depth outline of the algorithm is given in Ghilain et al. (2011) and Ghilain et al. (2012). Consequently, it was designed to run on the resolution of MSG-observations, but its capabilities at sub-kilometer scale were recently demonstrated (Barrios et al., 2020). For this study, the model was configured to run at kilometre-scale (i.e. the local scale corresponding to the footprint of eddy covariance measurements), using LAI from SPOTV and PROBAV and soil moisture from ERA5.

More recently, a LSA SAF GPP product was developed, based on the Monteith light use efficiency (LUE) concept (Martínez et al., 2020). This product is produced at the end of the LSA SAF pipeline, as it relies on several other LSA SAF products, such as ET, referenceET, LAI and FAPAR. The same formulation was adopted in the diagnostic model in our study, resulting

---

[1]https://www.eumetsat.int/about-us/satellite-application-facilities-safs





| Forcing | DiagMod | ISBA | ORCHIDEE |
|---|---|---|---|
| Air Temp (2m) | ERA5 | ERA5 | ERA5 |
| Specific Humidity | - | ERA5 | ERA5 |
| Wind | Tower | ERA5 | ERA5 |
| Wind Direction | - | ERA5 | ERA5 |
| Atmospheric pressure | ERA5 | ERA5 | ERA5 |
| Precipitation Rain | - | Tower | Tower |
| Precipitation Snow | - | Tower | Tower |
| Short Wave radiation | Tower | Tower | Tower |
| Long Wave radiation | Tower | Tower | Tower |
| $CO_2$ concentration | - | TRENDY | TRENDY |
| Soil Moisture | ERA5 | prognostic | prognostic |
| LAI | CGLS | prognostic | prognostic |
| FAPAR | CGLS | - | - |
| Soil Type | ECMWF | HWSD | FAO/USDA |
| Soil Layers | 4 | 14 | 12 |

**Table 1.** Source of forcing variables. Tower: Fluxtower observations from FLUXNET2015 dataset (Pastorello et al., 2020) and the ICOS '2018 drought initiative' dataset (Drought 2018 Team and ICOS Ecosystem Thematic Centre, 2019), TRENDY (Sitch et al., 2015, https://sites.exeter.ac.uk/trendy), CGLS: Copernicus Global Land Service (Camacho et al., 2013), ECMWF: soil texture used in the ECMWF Integrated Forecast System (https://apps.ecmwf.int/codes/grib/param-db?id=43), HWSD: harmonized world soil database (Nachtergaele et al., 2010), FAO/USDA: USDA texture map based on FAO digital Soil Map of the World (Reynolds et al., 2000)

in coherent surface fluxes.

Contrary to ISBA and ORCHIDEE, the calculations for LE and GPP in diagnostic model do not share parameters like stomatal resistance. Instead, the GPP calculations are coupled to LE by using the actual evapotranspiration as an input variable.

### 2.1.2 ISBA

Within the Surfex (SURFace Externalisée) land surface model, ISBA (Interactions between Soil, Biosphere and Atmosphere) is the component dedicated to modeling the exchange of water, energy and carbon fluxes between the soil-vegetation-snow continuum and the atmosphere (Masson et al., 2013; Le Moigne et al., 2018). In this case, a configuration of ISBA with interactive carbon cycling is used, i.e. ISBA-CC (Gibelin et al., 2008; Delire et al., 2020). The fluxes of water and carbon from the vegetation are coupled through the stomatal resistance. This shared parameter is calculated through the A-gs surface scheme, and largely depends on soil moisture stress and air temperature (Calvet et al., 2004). The parametrization for this scheme is based on plant traits derived from the TRY-database (Kattge et al., 2011; Delire et al., 2020).

The assimilation of carbon results in the evolution of LAI through a biomass allocation scheme. The growth and senescence





of leaves is purely photosynthesis-driven. The biomass reservoirs are coupled to a soil organic matter module to calculate the respiration terms.

The simulations with ISBA were performed on the Surfex v8.1 platform[2]. The soil profile was discretized in 14 layers (up to 12 m depth), using a diffusion scheme for soil heat and water transfer and an exponential decrease of hydraulic conductivity through the profile. The nitrogen dilution scheme (Calvet and Soussana, 2001) and canopy radiation transfer scheme (Carrer et al., 2013) were enabled. In the forest patches, the energy fluxes were calculated with the recently developed multi-energy

balance scheme (MEB, Boone et al., 2017). Contrary to the standard soil-vegetation composite version of ISBA (which was used for the non-forest patches), MEB explicitly solves the transfer of mass and energy between the soil surface, the snowpack, the canopy and the atmosphere. At the time of this study, the combination of MEB and prognostic LAI modelling is still considered experimental (Le Moigne et al., 2018). A spin-up period of 3 years was sufficient to eliminate effects from the initial model state on the surface fluxes (respiration is not analysed in this study). ISBA was not coupled to a hydrological

model (e.g. CTRIP Decharme et al., 2019). Consequently, there was no lateral ground water flow or a water table, only free drainage at the bottom of the soil profile.

### 2.1.3   ORCHIDEE

ORCHIDEE is the land surface model of the Institut Pierre Simon Laplace (IPSL) earth system model, and was initially described in Krinner et al. (2005). We used the version prepared for the 6th Coupled Model Inter-comparison Project (CMIP6)

(Boucher et al., 2020; Cheruy et al., 2020).

The LAI is prognostic, and the phenology models used for the various plant functional types (PFT) are described in Botta et al. (2000) and MacBean et al. (2015). The canopy is discretized in layers of increasing thickness from the top to the bottom of the canopy. The incoming light is attenuated through the canopy following a Beer-Lambert extinction law. The photosynthesis is modelled at the leaf level following Farquhar et al. (1980) for C3 species and Collatz et al. (1992) for C4 species. The maxi-

mum carboxylation rate at 25°C is a PFT-dependent parameter. The maximum carboxylation rate varies with the temperature following Medlyn et al. (2002) and Kattge and Knorr (2007). A water stress function depending on soil moisture and root profile (Rosnay and Polcher, 1998) is applied to the maximum carboxylation rate, the stomatal and the mesophyll conductances. An analytical solution to the three equations linking $CO_2$ assimilation, stomatal conductance and $CO_2$ leaf intracellular concentration is computed following Yin and Struik (2009). The assimilation is then upscaled over the layers to calculate the

GPP.

A single-layer energy balance is computed per grid cell. LE is the weighted average of the snow sublimation, the soil evaporation, the canopy transpiration and the evaporation of foliage water; all these terms were initially computed following Ducoudré et al. (1993). The soil is now discretized over 2 meters into 11 layers of increasing thickness, and the hydrology scheme follows Richard's equation (De Rosnay et al., 2002; d'Orgeval et al., 2008). There is free drainage at the bottom. The soil

thermodynamics are described in Wang et al. (2016), and the snow scheme is detailed in Wang et al. (2013).

---

[2]https://www.umr-cnrm.fr/surfex/





## 2.2 Test sites

The performance of the models was evaluated at field-scale, using observations from flux towers. A selection of sites was made to ensure adequate data quality, homogeneous land cover and limited disturbance due to management, resulting in the 56 sites listed in Table 2. 33 of these sites are dominated by forest land cover, whereas 18 are dominated by herbaceous vege-

tation and 5 are crop sites (the models are configured to run without management practices). The data was collected from the FLUXNET2015 dataset (Pastorello et al., 2020) and the ICOS '2018 drought initiative' dataset (Drought 2018 Team and ICOS Ecosystem Thematic Centre, 2019), resulting in a total of 526 simulation years. Both data products had been pre-processed with the ONEFLUX processing pipeline (Pastorello et al., 2020). The test sites were classified per PFT, dominant vegetation type (forest, herbaceous or crop) and hydro-climatic biome (HCB; Papagiannopoulou et al., 2018).

In addition to the classification based on land cover and meteorology, the sites were classified in 'drought classes'. In the land surface models, the rootzone soil moisture modulates the stomatal conductance when it drops below field capacity (ISBA) or below 80% of the difference between field capacity and wilting point (ORCHIDEE). As a proxy for the drought stress, the fraction of the simulation time that the simulated soil moisture in the topsoil (0-7 cm) drops below this threshold was used. It was found that this was significantly (Wilcoxon $p < 0.05$) more frequent in ISBA (48% of the time, median value of all sites),

compared to ORCHIDEE (26% of the time). Significant differences persisted deeper in the soil profile, until 70 cm depth. Using this metric, the sites were classified in four classes, equal in size, going from least (class 1) to most drought stress (class 4) (see Table 2). This classification was based on the ISBA simulations, but a similar classification was obtained with ORCHIDEE (despite the differences in absolute values). The vegetation at sites with drought class 1 was mainly dominated by forest, whereas the drought class 4 sites were mostly occupied by herbaceous vegetation. No evident relation with the hydro-

climatic biomes was found.



| Code | Name | Country | Database | Start | End | PFT | HCB | Köppen | LE corr | Drought |
|------|------|---------|----------|-------|-----|-----|-----|--------|---------|---------|
| AR-Vir | Virasoro | Argentina | FLUXNET2015 | 2009 | 2013 | ENF | Trans_W | Cfa | - | 2 |
| AU-ASM | Alice Springs | Australia | FLUXNET2015 | 2009 | 2013 | SAV | SubTr_W | BSh | 0.09 | 4 |
| AU-Ade | Adelaide River | Australia | FLUXNET2015 | 2006 | 2009 | WSA | Trans_W | As | 0.29 | 3 |
| AU-Cpr | Calperum | Australia | FLUXNET2015 | 2009 | 2014 | SAV | Trans_W | BSk | 0.02 | 4 |
| AU-DaP | Daly River Savanna | Australia | FLUXNET2015 | 2006 | 2013 | GRA | Trans_E | As | 0.22 | 3 |
| AU-DaS | Daly River Cleared | Australia | FLUXNET2015 | 2007 | 2014 | SAV | Trans_E | Aw | 0.03 | 3 |
| AU-Dry | Dry River | Australia | FLUXNET2015 | 2007 | 2014 | SAV | Trans_E | As | 0.27 | 4 |
| AU-How | Howard Springs | Australia | FLUXNET2015 | 2000 | 2014 | WSA | Trans_E | As | 0.19 | 3 |
| AU-Stp | Sturt Plains | Australia | FLUXNET2015 | 2007 | 2014 | GRA | Trans_E | As | 0.08 | 4 |
| AU-Wac | Wallaby Creek | Australia | FLUXNET2015 | 2004 | 2008 | EBF | Trans_E | Cfb | 0.12 | 1 |
| AU-Wom | Wombat | Australia | FLUXNET2015 | 2009 | 2012 | EBF | Trans_W | Cfb | 0.27 | 2 |
| BE-Bra | Brasschaat | Belgium | ICOS Drought | 1995 | 2018 | MF | MidL_T | Cfb | 0.17 | 1 |
| BE-Lon | Lonzee | Belgium | ICOS Drought | 2003 | 2018 | CRO | MidL_T | Cfb | 0.48 | 4 |
| BE-Vie | Vielsalm | Belgium | ICOS Drought | 1995 | 2018 | MF | MidL_T | Cfb | -0.03 | 1 |
| BR-Sa3 | Santarem | Brazil | FLUXNET2015 | 2000 | 2005 | EBF | Tropic | Aw | 0.17 | 2 |
| CA-Gro | Ontario | Canada | FLUXNET2015 | 2003 | 2015 | MF | Bor_T | Dfb | 0.42 | 1 |
| CA-NS6 | UCI-1989 burn site | Canada | FLUXNET2015 | 2001 | 2006 | OSH | Bor_T | BSk | - | 3 |
| CA-SF2 | Saskatchewan | Canada | FLUXNET2015 | 2001 | 2006 | ENF | Bor_T | Dwc | 0.41 | 3 |
| CA-SF3 | Saskatchewan | Canada | FLUXNET2015 | 2001 | 2007 | OSH | Bor_T | Dwc | 0.35 | 2 |
| CG-Tch | Tchizalamou | Congo | FLUXNET2015 | 2005 | 2009 | SAV | | As | - | 2 |
| CH-Lae | Laegeren | Switzerland | ICOS Drought | 2003 | 2018 | MF | MidL_T | Dfb | - | 1 |
| CN-Din | Dinghushan | China | FLUXNET2015 | 2002 | 2005 | EBF | SubTr_E | Cwa | - | 3 |
| CZ-BK1 | Bily Kriz forest | Czech Rep. | ICOS Drought | 2003 | 2018 | ENF | MidL_T | Dfb | - | 1 |
| DE-Kli | Klingenberg | Germany | ICOS Drought | 2003 | 2018 | CRO | MidL_T | Dfb | 0.46 | 3 |
| DE-Obe | Oberbrenburg | Germany | ICOS Drought | 2007 | 2018 | ENF | MidL_T | Dfb | 0.21 | 1 |
| DE-RuS | Selhausen Juelich | Germany | ICOS Drought | 2010 | 2018 | CRO | MidL_T | Cfb | 0.47 | 4 |
| DE-Seh | Selhausen | Germany | FLUXNET2015 | 2006 | 2010 | CRO | MidL_T | Cfb | 0.14 | 4 |
| DE-Spw | Spreewald | Germany | FLUXNET2015 | 2009 | 2014 | WET | MidL_T | Cfb | - | 3 |
| DE-Tha | Tharandt | Germany | ICOS Drought | 1995 | 2018 | ENF | MidL_T | Dfb | 0.26 | 1 |
| FI-Hyy | Hyytiala | Finland | ICOS Drought | 1995 | 2018 | ENF | Bor_WT | Dfb | 0.03 | 1 |
| FI-Let | Lettosuo | Finland | ICOS Drought | 2009 | 2018 | ENF | Bor_WT | Dfb | -0.26 | 1 |
| FR-Fon | Fontainebleau | France | FLUXNET2015 | 2004 | 2014 | DBF | MidL_T | Cfb | - | 3 |
| FR-LBr | Le Bray | France | FLUXNET2015 | 1995 | 2008 | ENF | Trans_E | Cfb | 0.21 | 2 |
| FR-Pue | Puechabon | France | FLUXNET2015 | 1999 | 2014 | EBF | Trans_E | Csa | 0.42 | 2 |
| GF-Guy | Guyaflux | Fr. Guiana | FLUXNET2015 | 2004 | 2015 | EBF | Tropic | As | - | 2 |
| GH-Ank | Ankasa | Ghana | FLUXNET2015 | 1989 | 1989 | EBF | Tropic | Aw | 0.56 | 1 |
| IT-Cpz | Castelporziano | Italy | FLUXNET2015 | 1996 | 2009 | EBF | Trans_E | Csa | 0.07 | 2 |
| IT-Ro1 | Roccarespampani | Italy | FLUXNET2015 | 1999 | 2008 | DBF | Trans_E | Csa | - | 3 |
| IT-SRo | San Rossore | Italy | FLUXNET2015 | 1998 | 2012 | ENF | Trans_E | Csa | 0.37 | 2 |
| JP-MBF | Moshiri | Japan | FLUXNET2015 | 2002 | 2005 | DBF | Bor_T | Dfb | - | 1 |
| JP-SMF | Seto | Japan | FLUXNET2015 | 2001 | 2006 | MF | Trans_E | Cfa | - | 1 |
| MY-PSO | Pasoh | Malaysia | FLUXNET2015 | 2002 | 2009 | EBF | Tropic | Af | -0.01 | 1 |
| NL-Loo | Loobos | Netherlands | ICOS Drought | 1995 | 2018 | ENF | MidL_T | Cfb | 0.05 | 1 |
| PA-SPn | Sardinilla | Panama | FLUXNET2015 | 2007 | 2010 | DBF | Trans_E | Aw | - | 2 |
| RU-Che | Cherski | Russia | FLUXNET2015 | 2001 | 2005 | WET | Bor_E | Dwc | - | 4 |
| RU-Fyo | Fyodorovskoye | Russia | ICOS Drought | 1997 | 2018 | ENF | Bor_WT | Dfb | -0.18 | 3 |
| SD-Dem | Demokeya | Sudan | FLUXNET2015 | 2004 | 2009 | SAV | SubTr_W | Aw | 0.62 | 4 |
| US-ARM | Lamont | United States | FLUXNET2015 | 2003 | 2013 | CRO | MidL_W | Cfa | 0.19 | 4 |
| US-Ivo | Ivotuk | United States | FLUXNET2015 | 2004 | 2008 | WET | Bor_E | Dwc | -0.15 | 2 |
| US-Me6 | Metolius | United States | FLUXNET2015 | 2010 | 2015 | ENF | Trans_E | Dsb | 0.46 | 3 |
| US-SRC | Santa Rita Creosote | United States | FLUXNET2015 | 2008 | 2015 | OSH | Trans_E | BSh | 0.65 | 4 |
| US-SRG | Santa Rita Grassland | United States | FLUXNET2015 | 2008 | 2015 | GRA | Trans_E | BSh | 0.30 | 4 |
| US-SRM | Santa Rita Mesquite | United States | FLUXNET2015 | 2004 | 2015 | WSA | Trans_E | BSh | 0.26 | 4 |
| US-Sta | Saratoga | United States | FLUXNET2015 | 2005 | 2010 | OSH | MidL_W | Dfb | - | 2 |
| US-UMd | UMBS Disturbance | United States | FLUXNET2015 | 2007 | 2015 | DBF | Bor_T | Dfb | - | 3 |
| ZA-Kru | Skukuza | South Africa | FLUXNET2015 | 1999 | 2013 | SAV | Trans_W | Csa | 0.21 | 4 |

**Table 2.** Selection of 56 FLUXNET/ICOS sites used in this study. Classification by PFT, HCB (Boreal/Mid-Latitude/Transitional/Subtropical/Tropical+Energy/Water/Temperature-driven) and Köppen. LE corr: relative change of the mean LE flux after correction for energy balance closure (no value: correction not available). Drought: drought class, derived from ISBA simulations



Not all sites are equipped with soil moisture sensors, nor is there a standardized setup or post-processing for soil moisture in the FLUXNET or ICOS framework. Consequently, the validation of the simulated soil moisture and the sensitivity analysis were only performed for the sites with sensors. Furthermore, some sites were equipped with multiple sensors in the soil profile.

Here, only the median score of the sensors was used in the statistics (i.e. one score per site). For the validation, all sensors up to 2 m depth were used, whereas only the sensors up to 0.5 m depth (i.e. the shallow root zone) were used in the sensitivity analysis (though the impact on the results was minimal).

## 2.3 Validation

The simulated H and LE were validated with the observed daily mean fluxes from flux towers. The non-closure of the energy balance is a well-known issue in the eddy covariance observations (Cui and Chui, 2019). The turbulent fluxes in the FLUXNET and ICOS dataset were corrected for this, under the assumption that the measured Bowen ratio is correct (Pastorello et al., 2020). Due to missing observations of the ground heat flux, this correction was not possible for all sites. The validation of H and LE was only performed for the sites where all fluxes were available. The mean correction of LE of each site is listed in

Table 2.

Similarly, the simulated GPP was validated with the FLUXNET/ICOS GPP data. The net ecosystem exchange (NEE) observed at the fluxtower was partitioned into its ecosystem respiration (RECO) and GPP components using the daytime fluxes and constant friction velocity (USTAR) threshold method (Pastorello et al., 2020). Only data with a quality flag indicating good quality (1) or better was used in this analysis.

An important key to the feedback mechanism between the surface fluxes is the LAI. The simulated LAI from ISBA-CC and ORCHIDEE was validated using the remote sensed LAI from the European Copernicus Global Land Service [3]. The LAI data product used here is derived from SPOT-VGT and PROBA-V satellite data, it has a spatial resolution of 1 km and a temporal resolution of 10 days (Camacho et al., 2013). The sites were selected to be fairly homogeneous within the footprint area, and the observed LAI is assumed to be representative for the direct surroundings of the eddy covariance stations.

The simulated soil moisture profiles of ISBA and ORCHIDEE, and the ERA5 soil moisture (used in DiagMod), were validated where possible. To reduce biases caused by different soil physical properties of the soil profiles or differences in scale between models and observations, the observed and simulated volumetric soil moisture ($\theta$) was converted to the effective saturation ($S_e$) as follows:

$$S_e = \frac{\theta - \theta_{min}}{\theta_{max} - \theta_{min}} \tag{1}$$

where $\theta_{min}$ and $\theta_{max}$ were assumed to be the 5th and 95th percentile of the observed soil moisture in a site for the observations, or the residual and saturated water content for the simulations.

For H, LE, GPP, LAI and $S_e$, the classical validation indices are calculated: mean error (ME), root mean square error (RMSE),

---

[3]http://land.copernicus.eu/global/





Pearson correlation (r) and Nash-Sutcliffe model efficiency (NS). They were calculated as in Equations 2 - 5, in which $y^*$ and $y^o$ are the predicted and observed values, $\overline{y}$ the mean of $y$ and $n_o$ the number of observations:

$$\text{ME} = \frac{\sum^{n_o}(y^* - y^o)}{n_o} \tag{2}$$

$$\text{RMSE} = \sqrt{\frac{\sum^{n_o}(y^* - y^o)^2}{n_o}} \tag{3}$$

$$r = \frac{\sum^{n_o}(y^* - \overline{y^*})(y^o - \overline{y^o})}{\sqrt{\sum^{n_o}(y^* - \overline{y^*})^2 \sum^{n_o}(y^o - \overline{y^o})^2}} \tag{4}$$

$$\text{NS} = 1 - \frac{\sum(y^* - y^o)^2}{\sum(y^o - \overline{y^o})^2} \tag{5}$$

Taylor diagrams were constructed using the Pearson correlation ($r$) and standard deviation ($\sigma$) of the observed and simulated
variables. The validation was performed using the daily totals/averages.

Furthermore, the same analysis was also performed on the anomalies to the mean annual cycles, to isolate the capability of the models to capture seasonal variability. The validation indices of the seasonal anomalies have the subscript $_{\text{ANOM}}$, e.g. $\text{NS}_{\text{ANOM}}$. Significant differences between the models were evaluated with the Wilcoxon signed-rank test, and the significance of the PFT and HCB to classify the model performances was evaluated with the Kruskal-Wallis H-test.


### 2.3.1 Sensitivity to Soil moisture and LAI

To assess the sensitivity of the fluxes to the state variables ($S_e$ and LAI), the slope of the seasonal anomalies of the fluxes against the anomalies of the state variables was determined. This analysis was performed for the observations and the simulations, and compared. Note that the linear slope was used here, though a linear response is not necessarily expected (e.g. the
response to soil moisture anomalies depend on a wet/dry regime). The goal of this analysis was to investigate whether LSM are capable of reproducing a similar relationship as found in the observations. Significant differences between the models were evaluated with the Wilcoxon signed-rank test.

To evaluate whether errors in the state variables result in errors in the surface fluxes (or vice versa), the spearman rank correlation between both was calculated. Since Copernicus LAI was the reference LAI, this analysis was not possible for LAI in
DiagMod.





### 2.3.2 Phenology

The detection of the start, maximum and end of the seasonal cycle (SOS, MOS and EOS) was achieved by applying a smoothing operation (20 day rolling mean), followed by a threshold procedure (Maleki et al., 2020). In this threshold procedure, the

minima and maxima were used to delineate the growing and senescent phase of the season. MOS was defined as the date when the maximum of the season is reached, SOS and EOS were defined at the date where the growing or senescent phase crosses the threshold value $T$. $T$ was calculated for each growing or senescent phase as $T = P_5 + 0.2(P_{95} - P_5)$, where $P_5$ and $P_{95}$ are the 5th and 95th percentile. This procedure was performed for the simulated and observed LE, GPP and LAI.

### 2.3.3 Model dynamics

To compare the model dynamics, the simulated LE flux partitioning, water balance and water use efficiency were evaluated as well. Direct observations of the LE flux partitioning were not available, but it is possible to extract the Transpiration component from the total LE flux, using the underlying Water Use Efficiency (uWUE) method (Zhou et al., 2016; Nelson et al., 2020).

## 3 Results

### 3.1 Validation surface fluxes: LE and GPP

The bias (ME) and accuracy (RMSE) of the simulated LE and GPP are shown in Fig. 2, together with Taylor diagrams of the simulated fluxes and their seasonal anomalies. It was evident that the inter-site variability of the model performance is much larger than the inter-model variability. In terms of bias and accuracy, the differences between the models were relatively limited. All models suffered a substantial underestimation of LE, whereas the overall bias in GPP was relatively small. Significant differences (Wilcoxon p<0.05) were found in the bias of GPP between DiagMod (overestimation) and ISBA (underestimation),

and the simulated LE was significantly more accurate in ISBA, compared to ORCHIDEE.

Notably, no substantial bias was found in the simulated H of any model to compensate for the consistent bias in LE (results shown in supplementary material). In this study, the corrected fluxes from the FLUXNET/ICOS dataset were used as a reference. They were corrected to close the energy balance of the observations with the assumption that observed Bowen ratio was accurate. If the non-corrected fluxes were used instead, the bias in LE was reduced, but the simulated H was overestimated

(not shown here). This points at the significant uncertainty associated with the observed fluxes from eddy-covariance measurements. The estimated observation uncertainty of the turbulent fluxes (associated with random measurement errors and energy balance correction) had the same order of magnitude as the model errors.

The Taylor diagrams in Fig. 2 show that the average variability of the simulated LE and GPP was in fair agreement with the observations. After removal of the mean seasonal cycle, the performance of the models decreased ($r_{\text{ANOM}}$, NS$_{\text{ANOM}}$), but the

mean variability of the anomalies is reasonably accurate. In terms of $r$ and $r_{\text{ANOM}}$ of LE and GPP, ORCHIDEE was significantly outperformed by ISBA and DiagMod (Wilcoxon p<0.05). No significant differences were found between ISBA and





DiagMod.

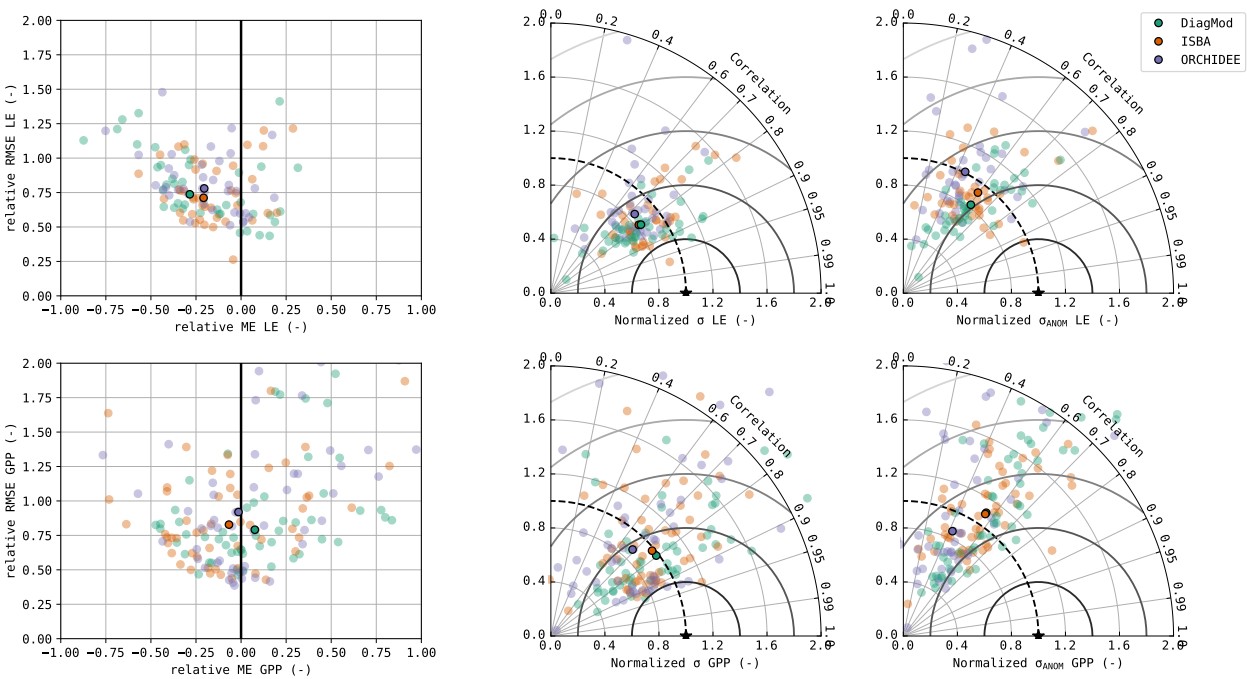

**Figure 2.** Accuracy plot (left), Taylor diagram (middle) and Taylor diagram of the seasonal anomalies (right) of the simulated daily mean LE (top) and GPP (bottom). The median performance is shown with the opaque markers.

The impact of the land cover type of the test site on the model performance is illustrated in Fig. 3. Here, the test sites are
classified by the dominant vegetation type. The NS and $NS_{ANOM}$ of the simulated LE was not significantly impacted in any of the models, whereas a significant influence (Kruskal $p<0.05$) was found on the quality of the simulated GPP in DiagMod and ORCHIDEE. The NS and $NS_{ANOM}$ of the simulated GPP in ORCHIDEE were significantly better (Mann-Whitney U $p <0.05$) for forest sites, compared to sites that were dominated by herbaceous vegetation. Inversely, the simulation of the seasonal GPP anomalies in DiagMod were significantly better in herbaceous test sites (Mann-Whitney U $p <0.05$). No significant impact
was found in the ISBA simulations. Notably, the differences between the models were most pronounced in the herbaceous sites (see also Table 3). Yet, despite its poorer performance in the herbaceous sites, ORCHIDEE simulated GPP in forest sites most accurately, compared to the other models.



|  |  | DiagMod |  |  |  | ISBA |  |  |  | ORCHIDEE |  |  |  |
|---|---|---|---|---|---|---|---|---|---|---|---|---|---|
|  |  |  | Forest | Herb | Crop |  | Forest | Herb | Crop |  | Forest | Herb | Crop |
| NS | LE | **0.46** | 0.56 | 0.34 | 0.35 | **0.49** | 0.42 | 0.58 | 0.64 | **0.39** | 0.40 | 0.36 | 0.43 |
|  | GPP | **0.37** | 0.33 | 0.32 | 0.44 | **0.31** | 0.44 | 0.10 | 0.27 | **0.15** | 0.46 | -0.79 | 0.27 |
|  | $S_e$ | **-0.01** | -0.11 | 0.37 | -0.70 | **-0.09** | -0.74 | 0.52 | -3.07 | **-0.37** | -0.71 | 0.14 | -2.89 |
|  | LAI |  |  |  |  | **-0.74** | -0.47 | -1.05 | -0.77 | **-1.56** | -1.12 | -3.91 | -0.83 |
| NS$_{\text{ANOM}}$ | LE | **0.28** | 0.26 | 0.27 | 0.34 | **0.21** | 0.15 | 0.32 | 0.34 | **-0.03** | -0.02 | -0.12 | 0.07 |
|  | GPP | **0.04** | -0.52 | 0.28 | 0.22 | **-0.07** | -0.01 | -0.14 | -0.19 | **-0.19** | 0.11 | -1.83 | -0.22 |
|  | $S_e$ | **0.09** | 0.01 | 0.36 | 0.03 | **0.14** | 0.06 | 0.47 | -0.14 | **0.06** | 0.02 | 0.43 | -0.12 |
|  | LAI |  |  |  |  | **-0.54** | -0.32 | -1.34 | -3.74 | **-0.50** | -0.03 | -4.06 | -2.68 |

**Table 3.** Nash-Sutcliffe model efficiency coefficient of LE, GPP, $S_e$ and LAI, and their seasonal anomalies. Median scores given for all sites, and grouped per dominant land cover type. The scores for the DiagMod $S_e$ are computed using the ERA5 $S_e$.

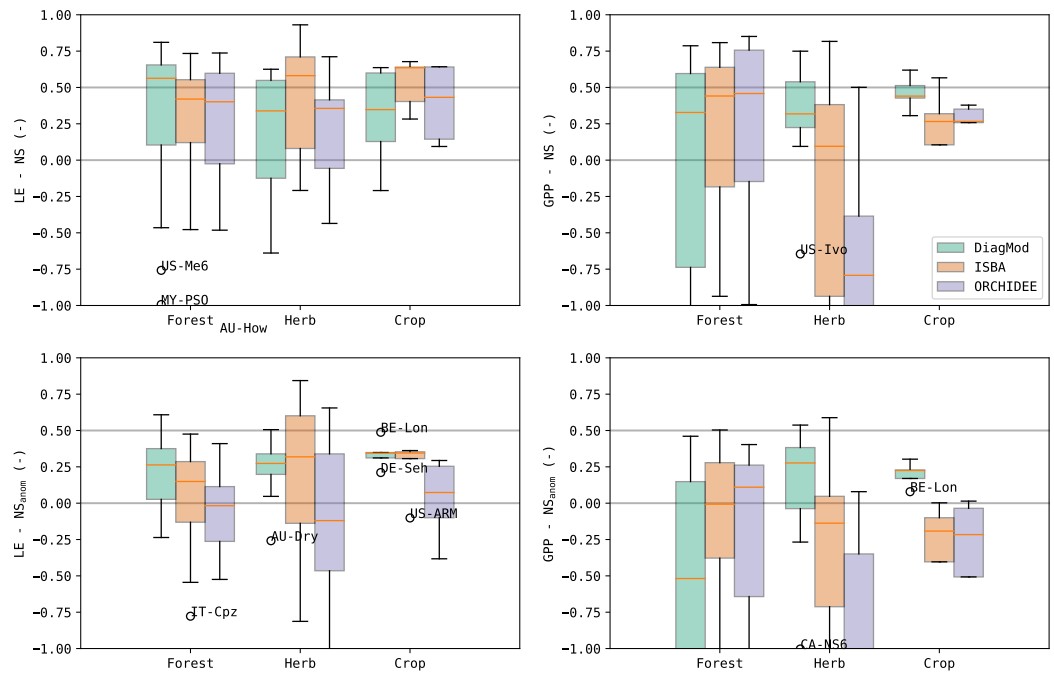

**Figure 3.** NS and NS$_{\text{ANOM}}$ of the simulated daily LE and GPP, grouped per land cover type

Similar results were found with the other validation indices. A more detailed breakdown of the results per PFT and HCB is
given in the supplement material. A significant impact of PFT and HCB on the NS of the simulated GPP (Kruskal p <0.05) was found in all models. This was contrasted by LE, where a significant impact of HCB (Kruskal p <0.05) was found only for ORCHIDEE.

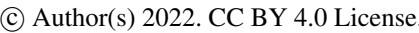



## 3.2  Validation state variables: Soil moisture and LAI

The validation results of $S_e$ and LAI are shown in Fig. 4. The soil moisture used in DiagMod was taken from ERA5, and tended to be overestimated compared to in situ observations, whereas an overall negative bias was found in ISBA and ORCHIDEE. No significant differences were found in $r$ between all models. The simulated variability of soil moisture was too low in all models, in particular for ORCHIDEE. Notably, ERA5 outperformed ISBA (p> 0.05) and ORCHIDEE (p< 0.05) in terms of accuracy, despite their use of in situ meteorological forcings (e.g. precipitation). ORCHIDEE performed significantly worse than the

other two models for all validation metrics (Wilcoxon $p < 0.05$). The highest correlation in the anomalies was simulated by ISBA.

Compared to the surface fluxes, the accuracy of the simulated soil moisture was substantially lower. The validation scores of $S_e$ are given in Table 3, separated per dominant land cover type. In all models, the simulated $S_e$ was significantly better for herbaceous sites, compared to forest sites. The herbaceous sites are generally found in a water-driven dryland climate, with

strong precipitation-driven anomalies.

Similarly, the prognostic LAI was also of poorer quality than the simulated surface fluxes. ISBA had a significantly better ME and RMSE than ORCHIDEE, but both models overestimated LAI and strongly underestimated its variability. In particular, the variability of LAI in the evergreen needleleaf forests was strongly underestimated in both models, as well as the variability of LAI in evergreen broadleaf forests in ORCHIDEE. Furthermore, both models obtained only a poor correlation, and achieved a

very poor correlation of the seasonal anomalies.

In both models, the simulated LAI for forest sites was better than for the herbaceous sites, though not significantly (p>0.05). The simulated anomalies were modelled significantly better (p<0.05) in forest sites than herbaceous sites (Table 3).




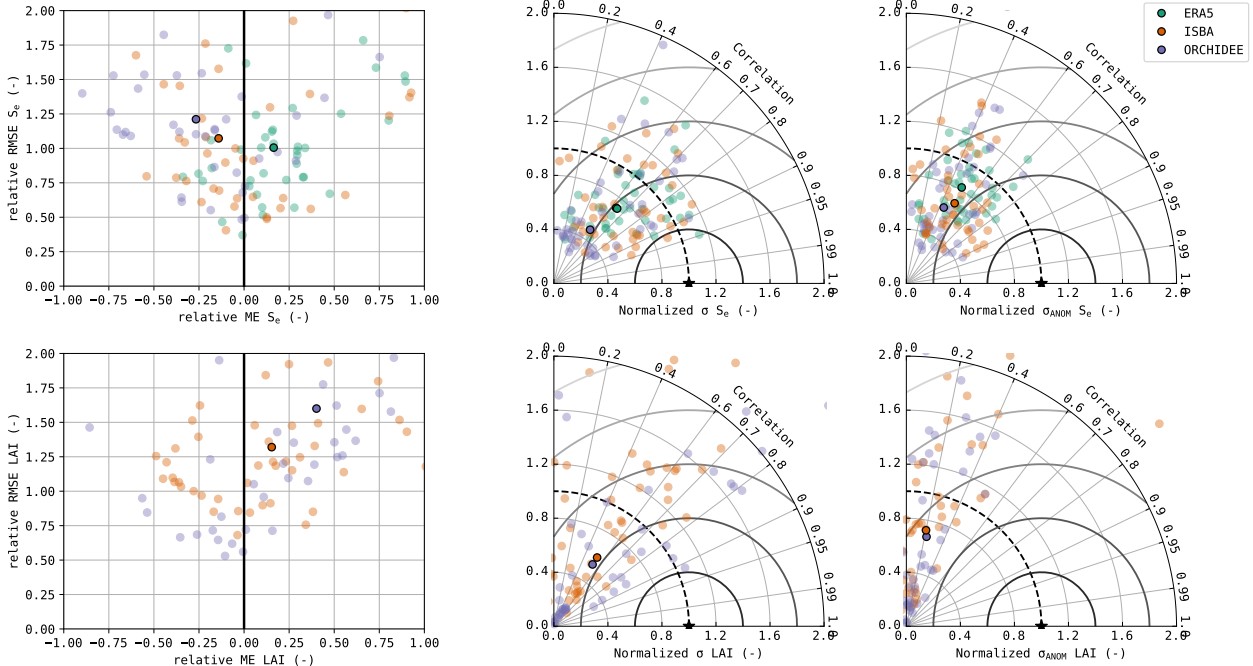

**Figure 4.** Accuracy plot (left), Taylor diagram (middle) and Taylor diagram of the seasonal anomalies (right) of $S_e$ (top) and LAI (bottom). The median performance is shown with the opaque markers.

## 3.3 Sensitivity of surface fluxes to soil moisture and LAI

The sensitivity of the fluxes to the soil moisture was strongly dependent on the land cover type, both in the observations and in the models (Fig. 5). A stronger response was found in the herbaceous sites, compared to the forest sites. ISBA and ORCHIDEE have a too high sensitivity to soil moisture, whereas the response in the diagnostic model was closer to that in the observations. Fig. 6, the same data is plotted, but classified per drought class. This illustrates the oversensitivity of ISBA and ORCHIDEE to drought stress. Despite their differences in implementation and parametrisation, a striking similarity in their sensitivity to drought was found, both for LE and GPP. The observations did not show an increase in sensitivity of GPP to $S_e$ in dryer sites. In the forest sites, the response of GPP to $S_e$ anomalies is counter-intuitively negative. This might indicate that soil moisture anomalies in forest sites were more dominated by wet anomalies, associated with rainfall events. These events coincide with a reduction in solar radiation, hence resulting in a negative GPP response. In herbaceous sites soils were generally drier, so the positive impact of the reduced drought stress after the rainfall event was more dominant, resulting in a positive response. This behaviour was mimicked well in the models.

The sensitivity of LE and GPP to LAI was generally higher in the herbaceous sites. Here, the models tended to underestimate the sensitivity to LAI. In the forest sites, the sensitivity was lower according to the observations. The modelled sensitivity of LE to LAI was reasonably accurate, whereas the sensitivity of GPP to LAI was too strong.





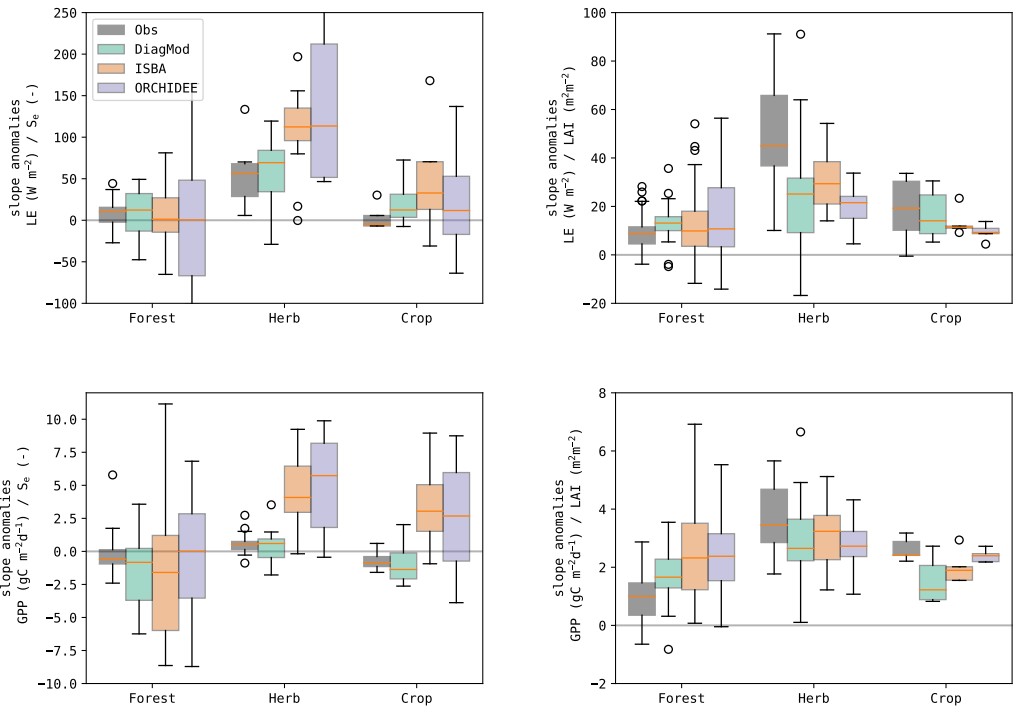

**Figure 5.** Boxplots of the Spearman Slope between the anomalies in the state variables ($S_e$ and LAI) and the fluxes (LE and GPP) in the test sites, grouped per dominant land cover

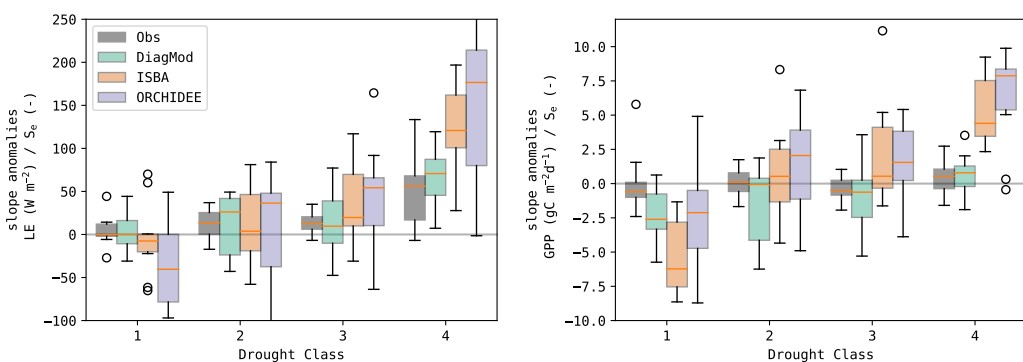

**Figure 6.** Boxplots of the Spearman Slope between the anomalies in the state variables ($S_e$ and LAI) and the fluxes (LE and GPP) in the test sites, grouped per drought stress class (1: least - 4: most frequent drought stress)





To evaluate the impact of the quality of $S_e$ and LAI on the simulated surface fluxes, the Spearman correlation of the errors in
the state variables and the fluxes was calculated (Fig. 7). It was found in both ISBA and ORCHIDEE that LAI had a stronger
error correlation to LE and GPP, compared to $S_e$. Grouped per dominant land cover type (Fig. 8), both models agree that the
error correlation between LAI and GPP was higher in the herbaceous sites, compared to the forest sites. Notably, this was not
the case for LAI-LE.

Furthermore, the errors in $S_e$ were strongest correlated to those in LE for all models. The highest error correlation was found
in DiagMod, where this was most pronounced for the herbaceous sites. In these sites, the $S_e$-GPP error correlation was also
the strongest for DiagMod, whereas no strong $S_e$-GPP error correlation was found in the other models.

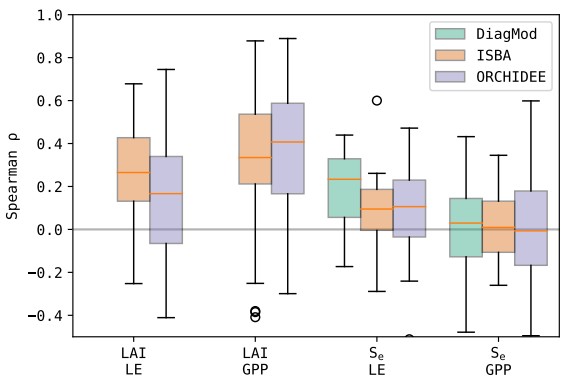

**Figure 7.** Boxplots of the Spearman correlation between the errors in the state variables ($S_e$ and LAI) and the fluxes (LE and GPP) in all test sites





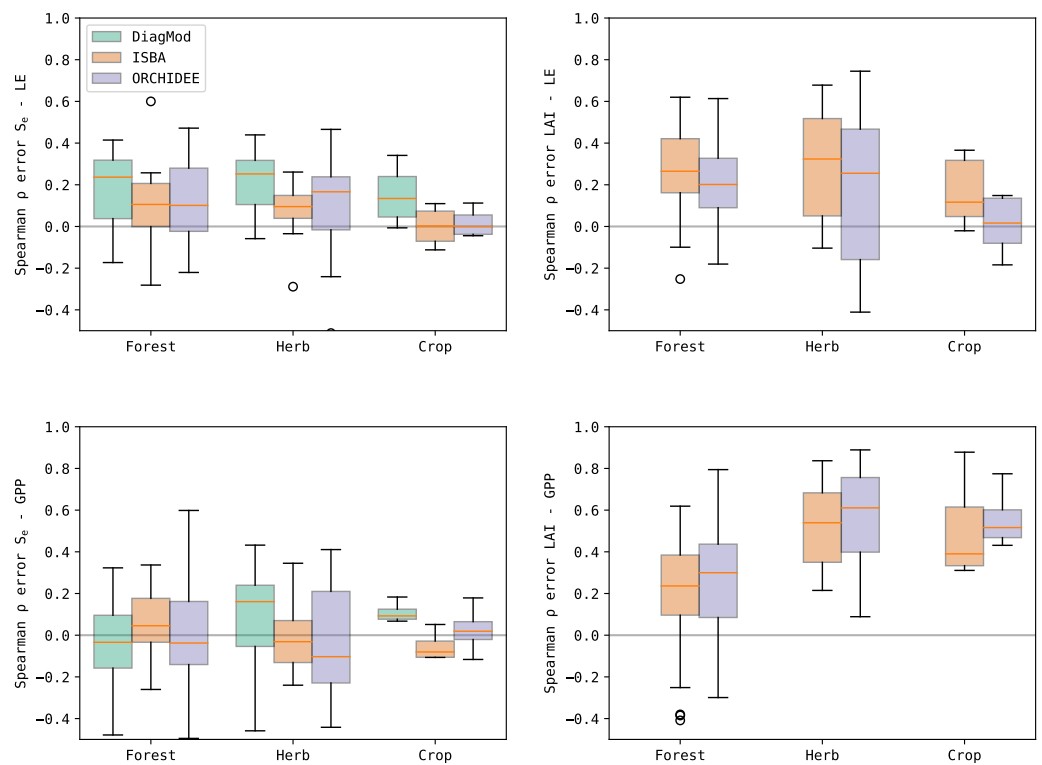

**Figure 8.** Boxplots of the Spearman correlation between the errors in the state variables ($S_e$ and LAI) and the fluxes (LE and GPP) in the test sites, grouped per dominant land cover

### 3.4 Phenology

The timing of the start, maximum and end of the seasonal cycle was validated for LE, GPP and LAI. Fig. 9 shows the boxplots

of the mean error in all sites. In all models, the bias and accuracy of the seasonality of LE and GPP was comparable, whereas the leaf phenology (i.e. LAI) was poorer. The simulated phenology of LAI was delayed substantially, in particular in ISBA. This bias was most pronounced by the MOS, and to a lesser extent in EOS.

ISBA performed significantly worse than ORCHIDEE (Wilcoxon $p < 0.05$) for ME of MOS GPP and MOS LAI, and RMSE of MOS LAI. The prognostic LAI in both models tended to peak towards the end of the growing season, whereas the maximum

LAI was reached in the beginning of the season according to the observations. This is illustrated in Fig. 10, where the mean annual LE, GPP and LAI cycles of ENF and DBF sites are shown. The delayed GPP phenology in ISBA is a feedback effect of the delayed prognostic LAI. However, the effect is dampened, since GPP is largely driven by other meteorological forcings as well.

In forest sites, EOS of LAI tended to be simulated with the highest accuracy. The phenology of herbaceous sites had a higher

variability (Median standard deviation of EOS LAI in forest sites was 7.7 days, compared to 20.6 days in herbaceous sites),





which turned out to be challenging to capture for ISBA and ORCHIDEE. An example is shown for the Savanna sites in Fig. 11. DiagMod relied on the remote sensed LAI and was significantly more accurate than the prognostic models to capture EOS of LAI in herbaceous sites.

As the models were configured to run without dedicated management practices for the crop sites, EOS was estimated too late

due to the harvest practice (Fig. 11). Even in DiagMod, EOS of GPP was delayed.

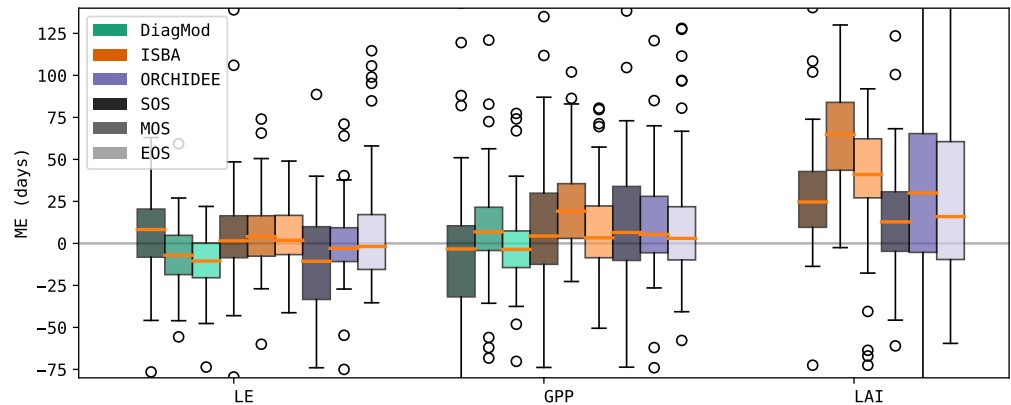

**Figure 9.** Mean error in the timing of the simulated seasonal cycle (Start, Max and End of season) for LE, GPP and LAI



**Figure 10.** Mean annual cycle for LE, GPP and LAI in all evergreen needleleaf forest (left) and deciduous broadleaf forest (right) sites, observed and simulated. Note: corrected LE observations were missing in all DBF sites.



**Figure 11.** Mean annual cycle for LE, GPP and LAI in all savanna (left) and crop (right) sites, observed and simulated





## 3.5 Water balance, WUE & LE partitioning

The water balance partitioning in ISBA and ORCHIDEE is shown in Fig. 12. In both models, the evapotranspiration fraction across PFT was similar, but substantial differences were found in the drainage and runoff in both models. Whereas nearly no
water was lost through runoff in the ISBA simulations, a substantial amount of runoff was simulated with ORCHIDEE. On the other hand, the drainage in ISBA was consistently larger than in ORCHIDEE. DiagMod does not compute a water balance, so could not be included in this comparison.

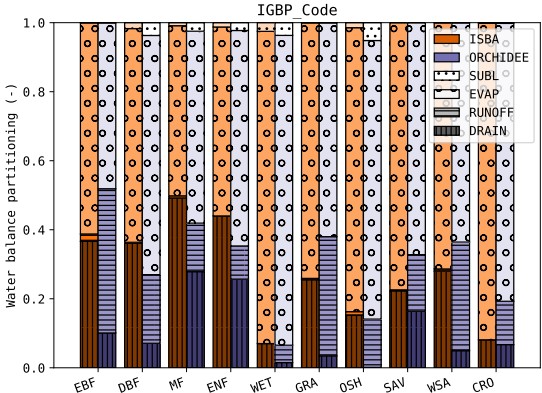

**Figure 12.** Average water balance partitioning (deep drainage, runoff, evapotranspiration and sublimination) per PFT class in ISBA and ORCHIDEE

Both models agreed that the largest fraction of LE is through transpiration of the vegetation (Fig 13). Aside from a few exceptions, T/ET in ORCHIDEE was larger than in ISBA. The median T/ET in ISBA (0.53) is lower than in ORCHIDEE
(0.68), and is closer to the values derived from the tower observations with the uWUE method (0.54). However, measurements by Lian et al. (2018) indicate that this is an underestimation, and suggest 0.62±0.06 as a global estimate.




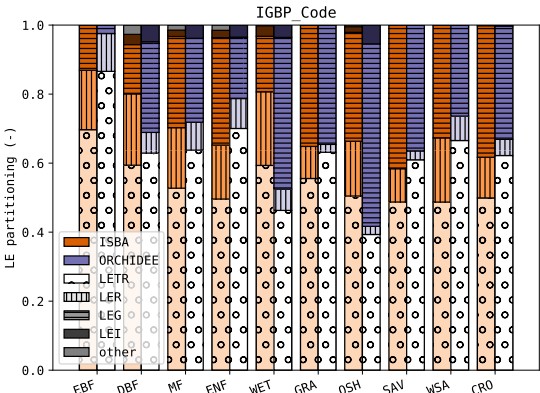

**Figure 13.** Average LE partitioning per PFT class in ISBA and ORCHIDEE. LETR: Transpiration, LER: intercept evaporation, LEG: soil evaporation, LEI: ice/snow evaporation and other, including evaporation from flooded surfaces.

When the observed average water use efficiency is plotted versus the average LE flux, a pattern emerges in which the sites are grouped per PFT (Fig. 14). A similar pattern was found in the ISBA simulations, but not in the ORCHIDEE simulations. The range in WUE across the test sites was much smaller in ORCHIDEE than in the observations.

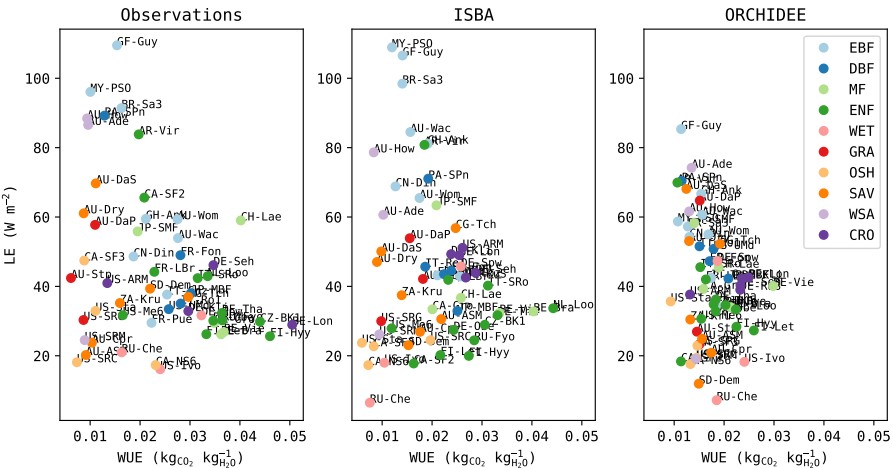

**Figure 14.** Median water use efficiency and LE in observations and simulations. Sites classified per PFT.

The difference in water use efficiency can be attributed to differences in the modelled plant physiology, or the amount of drought stress experienced by the vegetation. As mentioned above, the rootzone soil moisture dropped significantly more frequent below field capacity in ISBA, compared to ORCHIDEE.



## 4 Discussion

### 4.1 Model performance

The validation metrics of the three models were generally in agreement with previously performed local scale evaluations. Similar simulations with the diagnostic model were done in the validations reports of both the LSA SAF evapotranspiration and surface fluxes products (Ghilain et al., 2018) in one hand and the LSA SAF GPP product (Martínez et al., 2020) in the other hand. The accuracy and Pearson correlation obtained here were better than the ones previously reported. This can be attributed to the use of local forcings in this study, which are not used in the LSA SAF products. The weaker performance of

the algorithm for the sensible heat flux was also identified by Ghilain et al. (2018).

The GPP product is a recent addition to the ensemble of LSA SAF MSG products. It was demonstrated to outperform similar products which also rely on the Montheith light-use efficiency method (Martínez et al., 2020). Here, it was found perform consistently well for forest and herbaceous sites, and achieve a comparable model performance as ISBA.

In previous intercomparison studies at local scale (Balzarolo et al., 2014) or global scale (Friedlingstein et al., 2021), GPP was

simulated more accurately with ORCHIDEE than with ISBA, but this was not confirmed here. Since these studies, substantial improvements have been made to ISBA: introduction of the MEB scheme, parametrization update, diffuse multilayer soil scheme, etc. (Boone et al., 2017; Delire et al., 2020). The introduction of the MEB scheme for forests on the energy fluxes was evaluated in-depth by Napoly et al. (2017) at local scale (though prognostic LAI was not included in that study). Substantial improvements to G and H were reported, thanks to the addition of an insulating litter layer. The introduction of the MEB

scheme improved the mechanistic representation of the canopy, and issues due to a shared roughness length of the vegetation and bare soil in the composite scheme were circumvented. Our findings agree with that outcome, but the bias we found for LE, is not in agreement with previous findings.

### 4.1.1 Observation uncertainties

With the emergence of freely available data from eddy covariance network, the use of local datasets is an increasingly standardized approach to evaluate the performance of land surface models (Balzarolo et al., 2014; Napoly et al., 2017; Williams et al., 2020; Chen et al., 2018; Joetzjer et al., 2015). However, the eddy covariance observations notoriously suffer from substantial biases and non-closure of the energy balance (Foken, 2008; Mauder et al., 2020). The non-closure of the energy balance is attributed to 1) large advective fluxes caused by surface, 2) systematic measurement errors due to mismatch in observation

footprint or inadequate sample rate, 3) thermal processes, such as heat storage or vegetation metabolism (Mauder et al., 2020; Chu et al., 2021; Liu et al., 2021). The test sites in this study were selected to have a relatively homogeneous landcover. Regardless, the resulting uncertainty in the observations was of the same order of magnitude as the model errors. The turbulent fluxes are typically underestimated, as is the GPP (Massman and Lee, 2002; Gao et al., 2019). Note that GPP is not corrected for this possible bias in the ONEFLUX processing pipeline (Pastorello et al., 2020). Furthermore, some studies have indicated

that the eddy covariance observations are closer to lysimeter data if the energy balance is closed by correcting LE only (Gebler





et al., 2015). Considering this, the negative bias of the simulated LE (and GPP) in this study could be even underestimated.

### 4.1.2 Forest vs Herbaceous

Generally, the differences between the accuracy of the simulated surface fluxes was most distinct in the sites dominated by
herbaceous vegetation. These sites have the most pronounced inter-annual variability, and seasonal anomalies are strongly
driven by precipitation events (Weber et al., 2009). This can be largely attributed to their natural occurrence in dryer climates
and shallower root system, compared to forests. The seasonal cycle of LAI in the herbaceous sites, and its variability, was
simulated poorly with the prognostic models. The error correlation analysis indicated that these errors were strongly related to
errors in the surface fluxes.

Contrary to the prognostic models, the diagnostic model performed consistently well for all types of land cover. Only the
seasonal variability of GPP in forest sites was simulated less accurately than with the prognostic models. Whereas the remote-
sensed observations adequately captured this variability for the herbaceous sites, they seemed to fall short for the forest sites.

### 4.2 Interactions

LAI and soil moisture are two key variables in the interaction between water, energy and vegetation. Though our understanding
of the involved processes at leaf-level scale is advanced, it remains challenging to scale these relations to the canopy level. This
was illustrated by erroneous sensitivity of the models to LAI and soil moisture. As in previous studies, the sensitivity of LE and
GPP to soil moisture was generally overestimated (Piao et al., 2013; Huang et al., 2016) in ISBA and ORCHIDEE, whereas
the diagnostic model represented the observed sensitivity relatively well.

The interplay between LE and LAI was analysed in detail by Forzieri et al. (2018) and (Forzieri et al., 2020). The estimated
global sensitivity of LE to LAI ($3.66 \pm 0.45\,\mathrm{W\,m^{-2}}\,/\,\mathrm{m^2\,m^{-2}}$, according to Forzieri et al., 2020) is lower than the one reported
here, but the applied methodology was not the same. Contrary to our study, anomalies due to climatic drivers (i.e. precipitation,
temperature etc.) were factored out, resulting in a different response. The oversensitivity of LE to LAI in ORCHIDEE was
also not confirmed in our study. Still, in accordance to these studies, a stronger response between LE and LAI was found for
herbaceous/soil moisture supply-driven sites, compared to forest/demand-driven sites.

Despite the differences in their architecture and parametrization, ISBA and ORCHIDEE demonstrated similar behaviour in the
interaction between water, energy and vegetation. Comparable sensitivities and error correlations were found in both models,
indicating that they share common weaknesses in their implementation.

### 4.2.1 LAI

The errors in the surface fluxes were strongly correlated to errors in LAI for both prognostic models, even though their sen-
sitivity to LAI reflects the observed sensitivity reasonably well (compared to the sensitivity to soil moisture). This seemed to





indicate that the source of the errors in the fluxes lies in the feedback mechanism between GPP and LAI (i.e. biomass allocation and phenology), rather than in the forward link between GPP and LAI (i.e. photosynthesis and leaf to canopy upscaling).

The prognostic simulation of LAI in ISBA was introduced by Gibelin et al. (2006) and uses a fairly simple scheme. The latest
update was the revision of plant trait parameters according to the TRY database (Kattge et al., 2011; Delire et al., 2020). It has frequently been reported that the seasonal cycle of the simulated LAI in ISBA is delayed by a month or more (Lafont et al., 2012; Gibelin et al., 2008; Joetzjer et al., 2015). Delire et al. (2020) attributes this to the leaf longlivety parameter, and (Szczypta et al., 2014) mentions the vegetation undergrowth dynamics as a possible cause for the mismatch between remote sensed LAI and the prognostic LAI in LSM. However, the issue seems to be related to the architecture of the biomass allocation
scheme as well. The assimilated carbon is attributed to the leaf biomass pool first, from where it trickles down to the other pools. The consequence is that the simulated LAI in ISBA starts slow and builds up LAI until late in the second half of the season, whereas the observed seasonal LAI cycles reach a maximum in the first half of the growing season.

Improvements to prognostic LAI are required to increase the skill of the LSM to simulate surface fluxes. It is also demonstrated by the diagnostic model that a fairly simple model is capable of simulating the surface fluxes accurately, given accurate obser-
vations of LAI.

In that context, processes from ORCHIDEE and other LSM could be adopted to improve the fairly simple biomass allocations scheme in ISBA. The importance of non-structural carbohydrates to capture the leaf phenology in LSM is well-known, though rarely implemented (Asaadi et al., 2018). Fatichi et al. (2019) indicates that a full-grown canopy of a deciduous broadleaf forest contains approximately 30% of the total yearly assimilated carbon, yet it is grown in 1 month (1/5 - 1/6 of the growing
season). This rough simplification illustrates that reserve dynamics are essential to simulate the seasonal cycle of the vegetation accurately. Such dynamics are implemented in ORCHIDEE: once certain phenological criteria are fulfilled, the carbon in a reserve pool is allocated to leaf biomass to kickstart the phenological cycle. Still, despite the dedicated phenology module, non-structural carbohydrates reserve dynamics, and a more advanced leaf demography, simulating LAI remained challenging in ORCHIDEE. The timing of the phenological cycle was more accurate in ORCHIDEE, though the accuracy of the simulated
LAI was significantly poorer than ISBA. This was the case in particular for herbaceous vegetation. This tendency towards delayed phenology (and in particular a delayed leaf senescence) is found in most earth system models in CMIP5 and CMIP6 (Park and Jeong, 2021).

The discrepancy in complexity between the modelling of photosynthesis and that of the biomass allocation has been highlighted by several authors (Fatichi et al., 2016; Friend et al., 2019), though the main challenge lies in the parametrization of
those processes. The allocation of carbon in terrestrial vegetation is an important knowledge gap, hindering the advancement of earth system models.

Finally, there are several important differences between the remote sensed vegetation and the idealized vegetation in the models which need to be recognized when comparing both. Firstly, the role of the understory has a well-known impact on the remote sensed LAI (Camacho et al., 2013), whereas the LSM do not consider the separate evolution of an understory. This can result in
substantial differences in the seasonal cycle of LAI. This was illustrated by the differences in the simulations and observations of the LAI cycle at ENF sites. Continuous in situ LAI observations with hemispherical photography in ENF sites are rare,





but Rautiainen et al. (2012) reported that the effective canopy LAI (including non-green foliage) in FI-Hyy (boreal ENF site) remained constant from June till mid-September. This in agreement with the flat LAI cycle for ENF in ORCHIDEE, but is in contrast with the remote sensed LAI and the prognostic LAI in ISBA. In an empirical model based on in situ observations for

the FR-LBr site, LAI demonstrated a seasonal cycle. The understory was responsible for most of the seasonal variation, and 30% of the LAI was attributed to the understory during the summer (Rivalland, 2003). The seasonal cycle in the remote sensed LAI seems exaggerated (ranging between $1\,\mathrm{m^2\,m^{-2}}$ in winter and $4\,\mathrm{m^2\,m^{-2}}$ in summer).However, considering the understory and seasonal variation in needleleaf greenness (Seyednasrollah et al., 2021), assuming a flat LAI does not seem accurate nether, in the context of simulating GPP.

Which brings up a second issue: the remote sensed LAI is the 'green' LAI, i.e. photosynthetically active leaves (Camacho et al., 2013). Whereas LAI in LSM is a key variable which wears many hats. A single LAI variable is used to represent the role of leaves in several processes (photosynthesis, interception, canopy radiation transfer, surface roughness, etc.), in which the greenness of the canopy is not always important. These discrepancies contribute to the mismatch between LAI in the observations and the models. Addressing them might further advance the representation of vegetation in LSM.

**4.2.2 Soil moisture**

A significant difference between ISBA and ORCHIDEE is found in the simulated soil moisture dynamics, the water partitioning and the water use efficiency. The simulated WUE in ISBA was in fair agreement with what is deduced from the eddy covariance observations. In contrast, the WUE in ORCHIDEE had a much narrower range. The comparison of the LE partitioning learns also that a larger fraction of the water was transpired in ORCHIDEE, compared to ISBA. The differences in WUE

and flux partitioning could be attributed to differences in the simulated plant physiology, or to the quality of the simulated soil water content. The variability of the simulated water content in ORCHIDEE was strongly underestimated, and the vegetation experienced significantly less drought stress in ORCHIDEE. It is likely that this translated to a low variability in WUE as well. Furthermore, a substantial part of the precipitation was lost as surface runoff, compared to ISBA. Though we did not have validation data to evaluate the water partitioning, it seems that the simulations of ORCHIDEE could be improved significantly

by addressing the soil moisture dynamics.

Overall, the accurate simulation of soil moisture and water infiltration is a challenge (perhaps one of the main challenges) in land surface models (Vereecken et al., 2019). The poor quality of the simulated soil moisture is also evident in this study, despite the use of the multi-layer diffuse water transport scheme. In this study, the superior simulation of soil moisture in ISBA contributes to the good performance in simulating the surface fluxes, in particular for sites with herbaceous vegetation and

water-driven climate. The soil physical parameters are determined using a global pedotransfer model, using only texture as input. New, advanced pedotransfer functions have emerged in recent years, using not only texture, but also climatology and land use as predictors (Gupta et al., 2021). As soil moisture is at the basis of many processes in LSM, incorporating these PTF seems the logical new step forward in LSM (Fatichi et al., 2020).

The local scale simulations in this study were not coupled to a hydrological model, thus ground water dynamics were lacking.

Though only a limited effect of capillary rise was found in studies with a coupled groundwater hydrology, the impact can





be non-negligible for forest ecosystems with a deep root system (Decharme et al., 2019; MacBean et al., 2020). The further development of ground water dynamics in LSM is indispensable for the accurate coupling of energy, water and carbon in forest vegetation and its response to severe drought events.

Several efforts have already explored the potential of improving soil moisture dynamics in LSM. Substantial improvements to

soil moisture have indeed been obtained by calibrating the pedotransfer functions, or soil physical parameters. Yet, the impact thereof on the surface fluxes has been found to be relatively limited (Pinnington et al., 2021), or in some cases even negative (Raoult et al., 2021). Though many parameters in ISBA and ORCHIDEE are derived from databases (Delire et al., 2020), the LSM have been calibrated to produce accurate surface fluxes using (amongst others) eddy covariance observations. The limited accuracy of the soil moisture dynamics might have been overcompensated in the resulting parametrization (Raoult et al., 2021).

The oversensitivity to drought stress in ISBA and ORCHIDEE is possibly an illustration of this. Improvements to the intricate network of gears under the hood of LSM are a delicate matter. Addressing the soil moisture dynamics should go hand in hand with corrections to the oversensitivity to drought stress.

## 5 Conclusions

Three land surface models were compared at local scale, using identical meteorological forcing and prescribed land cover. The

goal was to evaluate their skill to simulate surface fluxes (LE and GPP), as well as their simulated interaction between water, energy and vegetation. It was found that the diagnostic model (based on LSA SAF algorithms) performed consistently well for all land covers. The prognostic models (ISBA and ORCHIDEE) performed similarly well for the forest sites, but the simulations for herbaceous sites revealed some important shortcomings. The sensitivity analysis demonstrated that both models overestimate the sensitivity to drought stress, which was occurring most frequently in herbaceous sites. On the other hand,

the error analysis showed that errors in the prognostic LAI (and not soil moisture) were the dominant source of errors for LE and GPP in ISBA and ORCHIDEE. Given the acceptable sensitivity to LAI, the source of these errors is likely found in the feedback mechanism between GPP and LAI. Compared to observations, the simulated phenological cycle in both models was delayed and failed to capture the observed seasonal variability. Improvements in the leaf phenology and biomass allocation scheme are required to improve the simulated surface fluxes.

The analysis here demonstrated key strengths and weaknesses of each LSM. Most notably, we showed that ISBA and OR-CHIDEE shared key deficiencies concerning the coupling of the water, energy and vegetation, despite their differences in architecture and parametrization. Improving the feedback between GPP and LAI, the soil moisture dynamics, and the oversensitivity to drought might advance the performance of these LSM significantly.

*Code and data availability.* The scripts and datasets used in this study are freely available upon request to the authors.





## Appendix

## Test sites

The sites were classified according to plant functional type (PFT) and Hydro-climatic biome (HCB; Papagiannopoulou et al., 2018). The distribution of the sites according to this classification is shown in Fig. A1.

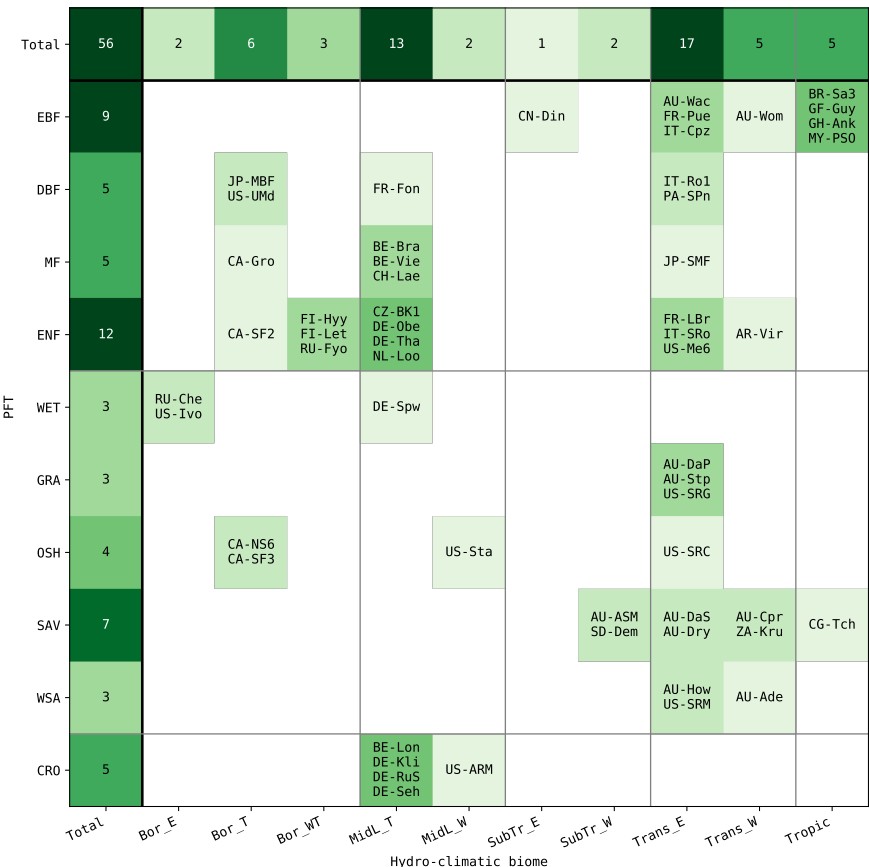

**Figure A1.** Overview of the selected FLUXNET sites, classified according to PFT and HCB. The colorscale indicates the number of sites in each class.





**530 Validation results per PFT and HCB**

**Figure A2.** Validation indices for H: a) ME, b) Nash-Sutcliffe, c) Nash-Sutcliffe for the seasonal anomalies

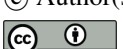



**Figure A3.** Validation indices for LE: a) ME, b) Nash-Sutcliffe, c) Nash-Sutcliffe for the seasonal anomalies





**Figure A4.** Validation indices for GPP: a) ME, b) Nash-Sutcliffe, c) Nash-Sutcliffe for the seasonal anomalies



## Phenology



**Figure A5.** ME of SOS, MOS and EOS for GPP




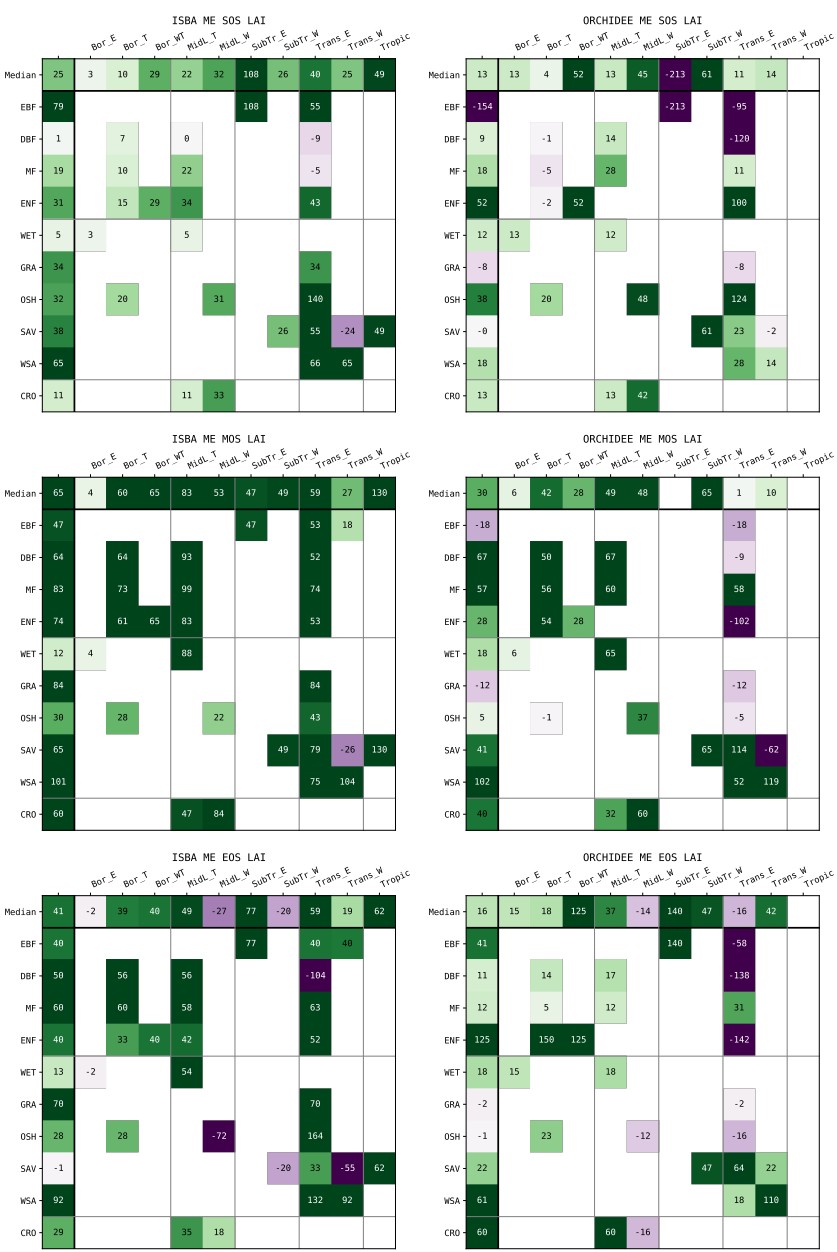

**Figure A6.** ME of SOS, MOS and EOS for LAI





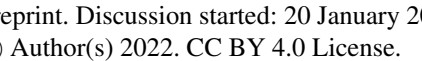

**Figure A7.** RMSE of SOS, MOS and EOS for GPP





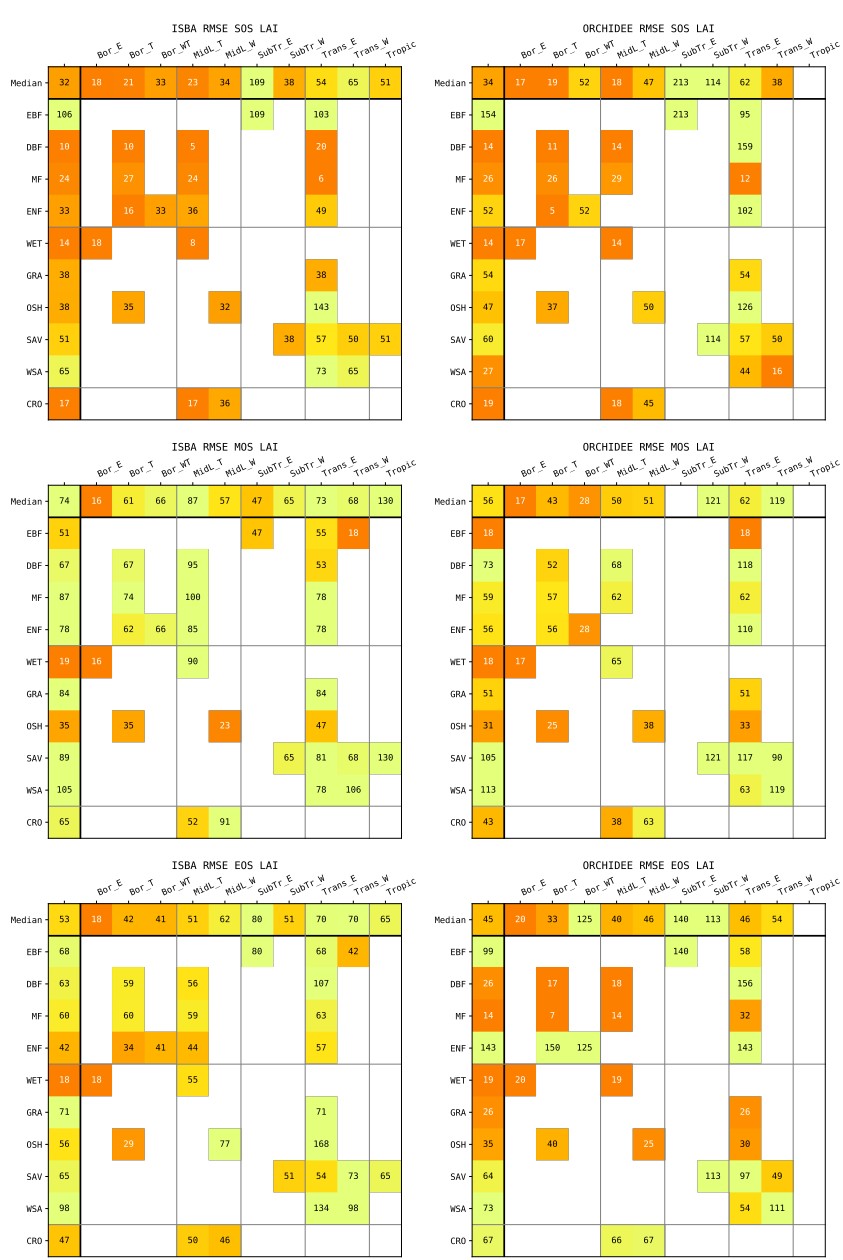

**Figure A8.** RMSE of SOS, MOS and EOS for LAI



*Author contributions.* JDP: Conceptualization, Investigation, Analysis, Writing - original draft preparation; MB and LL: Investigation, Analysis, Writing - review & editing;PC, AA, RH: Writing - review & editing; MB, FM, FGM: Supervision, Project administration, Writing - review & editing

*Competing interests.* The authors declare to have no competing interests

*Acknowledgements.* The research presented in this paper is funded by BELSPO (Belgian Science Policy Office) in the frame of the STEREO III programme – project ECOPROPHET (SR/00/334), and co-funded by EUMETSAT (LSA SAF program for CDOP-3) and the Belspo/ESA Prodex Program (PEA 4000110695). This work used eddy covariance data acquired and shared by the FLUXNET community, including these networks: AmeriFlux, AfriFlux, AsiaFlux, CarboAfrica, CarboEuropeIP, CarboItaly, CarboMont, ChinaFlux, Fluxnet-Canada, GreenGrass,
ICOS, KoFlux, LBA, NECC, OzFlux-TERN, TCOS-Siberia, and USCCC. The FLUXNET eddy covariance data processing and harmonization was carried out by the ICOS Ecosystem Thematic Center, AmeriFlux Management Project and Fluxdata project of FLUXNET, with the support of CDIAC, and the OzFlux, ChinaFlux and AsiaFlux offices. The assistance of the researchers who developed and maintain LSA-SAF, Surfex and ORCHIDEE was much appreciated. This work stands on the shoulders of the many who offer free, open source data, knowledge and tools. We thank Sci-hub for making scientific knowledge available to everyone.



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
