# Peer review of "Local scale evaluation of the simulated interactions between energy, water and vegetation in ISBA, ORCHIDEE and a diagnostic model."

_Biogeosciences, 2021_

## Author Comment (AC1)

**Rebuttal BG-2021-355**

Local scale evaluation of the simulated interactions between energy, water and
vegetation in land surface model

Jan De Pue, José Miguel Barrios, Liyang Liu, Philippe Ciais, Alirio Arboleda, Rafiq Hamdi,
Manuela Balzarolo, Fabienne Maignan, and Françoise Gellens-Meulenberghs
Handling Associate Editor: Ivonne Trebs

May 19, 2022
* * *
**JC Calvet**

Many thanks for this interesting work. This is an excellent contribution to the
continuous model intercomparison effort.

**Comment 1.1** — In order to allow the reproducibility of the simulations, it could
be useful adding a Supplement with model namelists together with a Table listing key
model parameters, their values, and how these values were determined.

> For ISBA and ORCHIDEE, the namelists used for this experiment are added as sup-
> plement material. Additionally, tables listing some of the key plant physiology and
> soil physical parameter values are given as well. The architecture of the diagnos-
> tic model differs too much from the prognostic models to allow comparison of the
> plant physiological parameters. They are not included in these tables.
> The following text has been added to section 3.1:
>
> > " An overview of some key plant physiology parameters and soil phys-
> > ical parameters is given in the supplement material, along with full
> > option namelists of the ISBA and ORCHIDEE runs to allow repro-
> > ducibility.
> > Most of the vegetation parameters in ISBA are derived from the TRY
> > plant traits database (Kattge et al., 2011; Delire et al., 2020). Pa-
> > rameters in ORCHIDEE are derived from the same database, but are
> > regularly calibrated using various data types, including satellite ob-
> > servations, in situ fluxes and surface concentrations (e.g. Kuppel et al.,
> > 2012, 2014; MacBean et al., 2015; Peylin et al., 2016). Kuppel et al.
> > (2014) used 78 FLUXNET sites to optimize parameters related to
> > the NEE (net ecosystem exchange) and LE fluxes (see their Table
> > S2). Hence, whereas the ORCHIDEE parameters were not optimized
> > using the specific dataset of this study, a part of it may have been
> > used formerly in this regard. Similarly, key parameters of the diag-
> > nostic model have been (indirectly) derived from the global network
> > of eddy covariance stations (Garbulsky et al., 2010; Martínez et al.,
> > 2020). "

| PFT | $V_{cmax}$ (mg m$^{-2}$s$^{-1}$) | | SLA (m$^2$kgC$^{-1}$) | |
|---|---|---|---|---|
| | ISBA | ORCH | ISBA | ORCH |
| Temp DBF | 2.60 | 1.14 | 15.4 | 26.0 |
| Boreal ENF | 2.80 | 1.02 | 5.0 | 9.3 |
| Trop EBF | 1.00 | 1.02 | 8.3 | 15.3 |
| C3 crop | 4.40 | 1.36 | 14.8 | 26.0 |
| C4 crop | 1.70 | 1.36 | 10.3 | 26.0 |
| irr. Crop | 3.40 | - | 10.3 | - |
| C3 grass | 3.40 | 1.14 | 14.0 | 42.0 |
| C4 grass | 1.70 | 1.14 | 5.7 | 41.0 |
| Wetlands | 3.40 | - | 14.0 | - |
| Trop DBF | 1.80 | 1.02 | 15.4 | 26.0 |
| Temp EBF | 2.60 | 0.91 | 8.3 | 20.0 |
| Temp ENF | 2.80 | 0.80 | 5.0 | 9.3 |
| Boreal DBF | 2.60 | 0.80 | 15.4 | 26.0 |
| Boreal DNF | 1.80 | 0.80 | 10.1 | 19.0 |
| Boreal Grass | 3.40 | - | 14.0 | - |
| Deciduous Shrub | 2.40 | - | 15.4 | - |

Table S-1: Key plant physiological parameters in ISBA and ORCHIDEE (for the conversion of $V_{cmax}$ to $g_m$ and $A_{mmax}$, see Delire et al. (2020). DBF = Deciduous Broadleaf Forest, ENF = Evergreen Needleleaf Forest, EBF = Evergreen Broadleaf Forest, DNF = Deciduous Needleleaf Forest.

| Texture class | $\theta_s$ | $\theta_r$ | $n$ | $\alpha$ | $K_s$ | $\theta_{fc}$ |
|---|---|---|---|---|---|---|
| | (m$^3$m$^{-3}$) | (m$^3$m$^{-3}$) | (-) | (10$^3$cm$^{-1}$) | (cm d$^{-1}$) | (m$^3$m$^{-3}$) |
| Sand | 0.43 | 0.05 | 2.68 | 14.5 | 7128.0 | 0.05 |
| Loamy Sand | 0.41 | 0.60 | 2.28 | 12.4 | 3501.6 | 0.07 |
| Sandy Loam | 0.41 | 0.07 | 1.89 | 7.5 | 1060.8 | 0.12 |
| Silt Loam | 0.45 | 0.07 | 1.41 | 2.0 | 108.0 | 0.24 |
| Silt | 0.46 | 0.03 | 1.37 | 1.6 | 60.0 | 0.26 |
| Loam | 0.43 | 0.08 | 1.56 | 3.6 | 249.6 | 0.17 |
| Sandy Clay Loam | 0.39 | 0.10 | 1.48 | 5.9 | 314.4 | 0.17 |
| Silty Clay Loam | 0.43 | 0.09 | 1.23 | 1.0 | 16.8 | 0.34 |
| Clay Loam | 0.41 | 0.10 | 1.31 | 1.9 | 62.4 | 0.27 |
| Sandy Clay | 0.38 | 0.10 | 1.23 | 2.7 | 28.8 | 0.27 |
| Silty Clay | 0.36 | 0.07 | 1.09 | 0.5 | 4.8 | 0.34 |
| Clay | 0.38 | 0.07 | 1.09 | 0.8 | 48.0 | 0.35 |

Table S-2: ORCHIDEE soil physical parameters for the Mualem-Van Genuchten model (derived from Carsel and Parrish (1988)), $\theta_{fc}$ is the water content at field capacity (not derived from the retention curve).

| Texture class | $\theta_s$ | $\Phi_s$ | $b$ | $K_s$ | $\theta_{fc}$ |
|---|---|---|---|---|---|
| | (m³m⁻³) | (cm) | (-) | (cm d⁻¹) | (m³m⁻³) |
| Sand | 0.40 | -12.10 | 4.05 | 1520.6 | 0.23 |
| Loamy Sand | 0.41 | -9.00 | 4.38 | 1350.7 | 0.24 |
| Sandy Loam | 0.44 | -21.80 | 4.90 | 299.5 | 0.32 |
| Silt Loam | 0.49 | -78.60 | 5.30 | 62.2 | 0.46 |
| Silt | 0.49 | -78.60 | 5.30 | 62.2 | 0.46 |
| Loam | 0.45 | -47.80 | 5.39 | 60.0 | 0.39 |
| Sandy Clay Loam | 0.42 | -29.90 | 7.12 | 54.4 | 0.35 |
| Silty Clay Loam | 0.48 | -35.60 | 7.75 | 14.7 | 0.42 |
| Clay Loam | 0.48 | -63.00 | 8.52 | 21.2 | 0.45 |
| Sandy Clay | 0.43 | -15.30 | 10.40 | 18.7 | 0.36 |
| Silty Clay | 0.49 | -49.00 | 10.40 | 8.9 | 0.46 |
| Clay | 0.48 | -40.50 | 11.40 | 11.1 | 0.45 |

Table S-3: ISBA soil physical parameters for the Campbell model (derived from Clapp and Hornberger (1978)),$\theta_{fc}$ is the water content at field capacity.

| Texture class | $\theta_s$ | $\theta_{fc}$ | $\theta_{wilt}$ |
|---|---|---|---|
| | (m³m⁻³) | (m³m⁻³) | (m³m⁻³) |
| Coarse | 0.40 | 0.24 | 0.06 |
| Medium | 0.44 | 0.35 | 0.15 |
| Medium fine | 0.43 | 0.38 | 0.13 |
| Fine | 0.52 | 0.45 | 0.28 |
| Very fine | 0.61 | 0.54 | 0.34 |
| Organic | 0.77 | 0.66 | 0.27 |
| Loamy | 0.47 | 0.32 | 0.17 |

Table S-4: DiagMod soil physical parameters, based on the Wösten et al. (1999) PTF.

**Comment 1.2** — It should be clarified whether any of the in situ datasets used in this study as a reference was involved in previous model parameter tunings.

The global network of eddy covariance station is directly or indirectly (via derived products, e.g. FluxCom) involved in the parametrization of land surface models. This is the case in particular for ORCHIDEE and the diagnostic model. The manuscripts has been revised to clarify this (see previous response).

**Comment 1.3** — Finally, how were model simulations initialized (e.g. initial root-zone soil moisture conditions)?

The initialization of ISBA is described in section 2.1.2, L133:

  " A spin-up period of 3 years was sufficient to eliminate effects from the initial model state on the surface fluxes (respiration is not analysed in this study). "

Concerning the initialization of ORCHIDEE, the following text was added to section 2.1.3:

> " To initialize the simulations, a first spin-up phase was performed, where we cycled over the available FLUXNET years for at least 45 years. This enables to reach an equilibrium for the above-ground biomass and the water stocks and fluxes, as an initial state for the transient simulation. "

The diagnostic model does not require any initialization.

**References**

R. F. Carsel and R. S. Parrish. Developing joint probability distributions of soil water retention characteristics. *Water resources research*, 24(5):755–769, 1988.

R. B. Clapp and G. M. Hornberger. Empirical equations for some soil hydraulic properties. *Water resources research*, 14(4):601–604, 1978.

C. Delire, R. Séférian, B. Decharme, R. Alkama, J.-C. Calvet, D. Carrer, A.-L. Gibelin, E. Joetzjer, X. Morel, M. Rocher, et al. The global land carbon cycle simulated with isba-ctrip: improvements over the last decade. *Journal of Advances in Modeling Earth Systems*, 12(9):e2019MS001886, 2020.

M. F. Garbulsky, J. Peñuelas, D. Papale, J. Ardö, M. L. Goulden, G. Kiely, A. D. Richardson, E. Rotenberg, E. M. Veenendaal, and I. Filella. Patterns and controls of the variability of radiation use efficiency and primary productivity across terrestrial ecosystems. *Global Ecology and Biogeography*, 19(2):253–267, 2010.

J. Kattge, S. Diaz, S. Lavorel, I. C. Prentice, P. Leadley, G. Bönisch, E. Garnier, M. Westoby, P. B. Reich, I. J. Wright, et al. Try–a global database of plant traits. *Global change biology*, 17(9):2905–2935, 2011.

S. Kuppel, P. Peylin, F. Chevallier, C. Bacour, F. Maignan, and A. Richardson. Constraining a global ecosystem model with multi-site eddy-covariance data. *Biogeosciences*, 9(10):3757–3776, 2012.

S. Kuppel, P. Peylin, F. Maignan, F. Chevallier, G. Kiely, L. Montagnani, and A. Cescatti. Model–data fusion across ecosystems: from multisite optimizations to global simulations. *Geoscientific Model Development*, 7(6):2581–2597, 2014.

N. MacBean, F. Maignan, P. Peylin, C. Bacour, F.-M. Bréon, and P. Ciais. Using satellite data to improve the leaf phenology of a global terrestrial biosphere model. *Biogeosciences*, 12(23):7185–7208, 2015.

B. Martínez, M. Gilabert, S. Sánchez-Ruiz, M. Campos-Taberner, F. García-Haro, C. Brümmer, A. Carrara, G. Feig, T. Grünwald, I. Mammarella, et al. Evaluation of the lsa-saf gross primary production product derived from seviri/msg data (mgpp). *ISPRS Journal of Photogrammetry and Remote Sensing*, 159: 220–236, 2020.

P. Peylin, C. Bacour, N. MacBean, S. Leonard, P. Rayner, S. Kuppel, E. Koffi, A. Kane, F. Maignan, F. Chevallier, et al. A new stepwise carbon cycle data assimilation system using multiple data streams to constrain the simulated land surface carbon cycle. *Geoscientific Model Development*, 9(9):3321–3346, 2016.

J. Wösten, A. Lilly, A. Nemes, and C. Le Bas. Development and use of a database of hydraulic properties of european soils. *Geoderma*, 90(3-4):169–185, 1999.

---

## Author Comment (AC2)

**Rebuttal BG-2021-355**

Local scale evaluation of the simulated interactions between energy, water and vegetation in land surface model

Jan De Pue, José Miguel Barrios, Liyang Liu, Philippe Ciais, Alirio Arboleda, Rafiq Hamdi, Manuela Balzarolo, Fabienne Maignan, and Françoise Gellens-Meulenberghs

Handling Associate Editor: Ivonne Trebs

May 19, 2022
* * *
**Anonymous referee 1**

This work presents a systematic point-scale evaluation of two prognostic land surface models (LSMs) and one observation-driven diagnostic model across several measurement sites of the FLUXNET monitoring network. The assessment of models' performance is focused on the simulated latent heat flux (LE), gross primary production (GPP), soil moisture, and leaf area index (LAI). Overall, this manuscript aims at disentangling the relative role of soil moisture and leaf area index in explaining the key models' weaknesses in the simulation of the land-atmosphere water-energy-carbon exchanges.

This work addresses a subject of interest for the broad audience of BG and it has the potential to shed additional light and provide guidance to the LSM modelling community on the simulation of water-energy-carbon interactions and involved feedback mechanisms. Having said that, I think there are substantial improvements to be implemented in the manuscript before making any consideration for publication.

I provide a more detailed list of comments below:

**Comment 2.1** — The **introduction** of the manuscript is **too weak**, lacking the clear definition of the unresolved research questions that are behind this work. Those provided between lines 70-74 are, in my opinion, not scientifically relevant to justify the publication of this manuscript. In addition, **a mere models vs observations comparison is not per se a strong objective**; see lines 79-84. On the other hand, if the main objective of the manuscript is included in the last sentence of the introduction ("Given the degree of coupling in the current LSM, we try to disentangle the relation between key facets of the terrestrial vegetation in a holistic way"), authors should put **more emphasis on this aspect and less on the evaluation** of the models' performance.

> As the reviewer points out, the manuscript balances between 2 aspects: model intercomparison and an analysis of the internal interactions. Rather than strictly reporting the model performances, we attempt to extend the analysis with some statistics to provide more insight in the origin of differences in the model performances.

The soil moisture and LAI were identified as primary explanatory variables. With the given analysis, we find that some meaningful conclusions can be derived, though -as the reviewer points out- a more extensive analysis would allow more solid statements. We agree with this remark, it would be very relevant to test more model configurations, in which we impose soil moisture from various sources (observed or prognostically simulated) to the models. However, there are a few limitations associated with the current implementation of the models:

- LAI and soil moisture cannot be imposed in ORCHIDEE

- only LAI can be imposed in ISBA

To address the remark of the reviewer, given the model limitations, we are investigating the following:

- On a sub-selection of 2 sites, the simulated soil moisture and LAI are improved by calibration of soil/vegetation related parameters in ORCHIDEE. The indirect effect on the surface fluxes will be evaluated.

- Similarly, ISBA simulations for these sites are repeated with imposed LAI and improved soil moisture simulation (by calibrating soil hydraulic parameters).

Additionally, as the reviewer suggests in point 2, the diagnostic model can be used as a vehicle to evaluate the prognostic variables from ISBA and ORCHIDEE. This interesting approach will be tested for all sites. However, to derive meaningful conclusions from this experiment, it needs to be verified that the diagnostic model reproduces similar surface fluxes as the prognostic models, using the prognostic soil moisture and LAI from those models.
This work is still ongoing, but will be undoubtedly a valuable addition to this manuscript.

**Comment 2.2** — I found the relative role "assigned" to the diagnostic model in the intercomparison exercise not fully clear and justified. Specifically, If the scope of the work is to compare the coherence with respect a LSM prognostic approach (see lines 73-74), authors should have structured their comparison in a different way. That is, they should have complemented the results of the **observation-driven (i.e., remote sensing for LAI and ERA5 for soil moisture) diagnostic model with those obtained assuming the output of the two LSMs** (i.e., soil moisture and LAI) as "observations". In this way authors should have been able to provide more stringent interpretations on the different models' performance and/or deficiency and eventually coherence.

As mentioned in the previous point, we agree that this would be a relevant experiment in this context. We will run the diagnostic model with prognostic soil moisture/LAI from ISBA/ORCHIDEE. The model configurations are listed in Table S-1. Some preliminary results are shown in Fig. S-1 and S-2.

| Label | LAI FAPAR | Soil Moisture |
|---|---|---|
| LAIcop_SWera | CGLS | ERA5 |
| LAIisb_SWera | ISBA | ERA5 |
| LAIcop_SWisb | CGLS | ISBA |
| LAIisb_SWisb | ISBA | ISBA |
| LAIorc_SWera | ORCHIDEE | ERA5 |
| LAIcop_SWorc | CGLS | ORCHIDEE |
| LAIorc_SWorc | ORCHIDEE | ORCHIDEE |

Table S-1: Model configurations. CGLS: Copernicus Global Land Service (Camacho et al., 2013), ERA5: Hersbach et al. (2020)

[Figure]

Figure S-1: Example of LAI and SWC time series (mean annual cycles, AU-DaP), used in the analysis with the diagnostic model. The dashed line in the soil moisture plot (layer 1) indicates the value of field capacity.

[Figure]

Figure S-2: Resulting LE and GPP mean annual cycles with different model configurations.

**Comment 2.3** — In a similar vein to the previous point, If the objective of the work is to understand how changes in the state variables (i.e., soil moisture and LAI) propagates to the surface fluxes (i.e., LE and GPP) and vice versa (see Figure 1), I think authors should **add additional LSM configurations** in the matrix of the conducted numerical experiments. That is, on top of the current ("free") configuration, each prognostic LSM should be also run using a **prescribed LAI** (i.e., from satellite products), prescribed soil moisture (if the two LSMs have this functionality), and with both LAI and soil moisture prescribed. I think this comprehensive numerical framework could

allow authors to get a "holistic" picture of the schematic shown in Figure 1 from three different LSMs.

> Indeed. As mentioned in Point 1, there are some limitations to the models, but a work-around is proposed, by calibrating the soil/vegetation parameters to improve the prognostic variables. However, due to the required time and energy for this test, we will limit ourselves to 2 sites, to demonstrate the impact of changing these variables. Performing this analysis for the full list of sites would require too much effort.

**Comment 2.4** — I would recommend **adding a table** summarizing the key differences between the models that could help interpreting/explaining results shown in the manuscript. In the current form, it is difficult to get a clear picture on what are the structural and parametrization features that could explain the different response in the three considered models.

> Such a table will be prepared for the next revision of the manuscript.

**Comment 2.5** — The **discussion section remains a bit too vague** in explaining the key reasons of the different model performances. I think the suggestions that I have provided in point 3 could help addressing this issue. As an example, the statement done between lines 438-440 could be validated using prescribed LAI values. The same apply to the sentence between lines 479-481. Overall, I think authors show make clear **what's the real objective** of their work. If the scope is to present a mere model **validation** exercise, the set of simulations presented in this study are sufficient, but they should try to **justify the novelty** of doing this in the introduction and better highlight the new insights gained by the large number of statistics. On the other way, if the purpose is to investigate how different LSMs resolve the **water-energy-carbon interactions**, I think the **current numerical setup provides not much information**.

> We do not fully agree with the reviewer. The current analysis of the sensitivities and error correlations allows to draw some relevant conclusions. However, we will follow the suggested approach to make a more in-depth analysis. The discussion will be adapted appropriately.

**References**

F. Camacho, J. Cernicharo, R. Lacaze, F. Baret, and M. Weiss. Geov1: Lai, fapar essential climate variables and fcover global time series capitalizing over existing products. part 2: Validation and intercomparison with reference products. *Remote Sensing of Environment*, 137:310–329, 2013.

H. Hersbach, B. Bell, P. Berrisford, S. Hirahara, A. Horányi, J. Muñoz-Sabater, J. Nicolas, C. Peubey, R. Radu, D. Schepers, et al. The era5 global reanalysis. *Quarterly Journal of the Royal Meteorological Society*, 146(730):1999–2049, 2020.

---

## Author Comment (AC3)

**Rebuttal BG-2021-355**

Local scale evaluation of the simulated interactions between energy, water and vegetation in land surface model

Jan De Pue, José Miguel Barrios, Liyang Liu, Philippe Ciais, Alirio Arboleda, Rafiq Hamdi, Manuela Balzarolo, Fabienne Maignan, and Françoise Gellens-Meulenberghs Handling Associate Editor: Ivonne Trebs

May 19, 2022

**Anonymous referee 2**

Three models (prognostic models ISBA and ORCHIDEE and a diagnostic model rooted in satellite remote sensing product development) were run on the meteorological and land surface forcing data of 56 eddy-covariance sites to compare their results to each other and to the site observations. Results were compared with multiple strategies (including bias vs. RMSE, Taylor diagrams, sensitivities between state variables per land use type, error correlations, phenology and seasonal cycles) and results were used to identify (where possible) or hyothesize (otherwise) weaknesses of the different models, also considering uncertainties in the observation data. Main findings include a better performance of the diagnostic as compared to teh prognostic models, a convergence in strengths and weaknesses between both prognostic models partly due to latest updates of the ISBA model, but also remaining differences, and recommendations on most important future improvements (in particular related to drought stress response, phenology and biomass allocation). The manuscript is very well written, methods and results are presented in clarity, the subject is relevant to Biogeosciences, and original among others in the sense that the newest version of ISBA and a comparatively new data product of teh eddy-covariance network are used.

Notwithstanding a lack of expertise on my side when it comes to internal details of the used models, which would ideally be addressed by other reviewers, I recommend the manuscript for publication after minor revisions suggested below. Detailed Comments:

**Comment 3.1** — Title: Consider adding "three" before land surface models, currently is lets readers easily think of a large multi-model study.

Agreed, the title was changed accordingly

**Comment 3.2** — Figure 1: Relation and feedbacks in the caption is no distinction that makes it easy to understand for the reader - relation could also be something purely empirical but here apparently you mean it as "more direct / stronger / first order" than the feedbacks. Check if this distinction is really needed and if, which other words

could stress it. Next line of the caption, from the arrow it would be better to write Soil moisture - LAI than vice versa. Figure itself: Would it make sense to add stomatal control somewhere in the middle? The way it is now it seems like LE and GPP are each controlled independently by soil moisture and LAI, such that the reader is almost wondering why there is not also an arrow between them.

The caption was corrected following the recommendations of the reviewer:

" First order relations (plain lines) and feedbacks (dashed lines) of the state variables and surface fluxes in prognostic LSM. The feedback mechanisms are not present in diagnostic models, and the Soil moisture-LAI relation (dotted line) occurs only in prognostic LSM with dedicated phenology schemes. "

Adding the stomatal control in the middle would make sense for the prognostic models. The soil moisture modulates the stomatal closure, which in turn affects LE and GPP. However, we choose not to display it as such, as this would not be an accurate representation of the mechanisms in the diagnostic model. Instead, we prefer to keep the more general schematic.

**Comment 3.3** — L97: "corrected manually to represent the tower footprint area" is somewhat unclear, could you be more specific?

This was rephrased:

"The land cover at each site was derived from ECOCLIMAP 2 (Faroux et al., 2013) and corrected manually if this was not representative for the tower footprint area (based on ICOS and FLUXNET metadata, and satellite imagery). "

**Comment 3.4** — L99: "linearly interpolated" refers to the ERA5 data being hourly and the tower observations mostly half-hourly?

Indeed, the text was revised:

" The forcing from ERA5 (hourly resolution) was linearly interpolated to match the 30 minute temporal resolution from the tower observations."

**Comment 3.5** — L140 & 159: Could the free drainage at the bottom explain the over-sensitivity to drought stress? (To be discussed not here)

Indeed, the lack of ground water dynamics might impact the results. This was referred to in the discussion section:

"The local scale simulations in this study were not coupled to a hydrological model, thus ground water dynamics were lacking. Though only a limited effect of capillary rise was found in studies with a coupled groundwater hydrology, the impact can be non-negligible for forest ecosystems with a deep root system (Decharme et al., 2019; MacBean et al., 2020). The further development of ground water dynamics in LSM is indispensable for the accurate coupling of energy, water and carbon in forest vegetation and its response to severe drought events."

The free drainage boundary condition might result in an overestimation of the occurrence of water-limited conditions. However, in the sensitivity analysis we only consider the relation between anomalies in the simulated soil moisture and the fluxes. In principle, the sensitivity of this analysis to the frequency of water-limited conditions should be limited. If there would be an issue caused by the free drainage boundary condition, it would be primarily visible in the overestimated frequency of water-limited conditions.

**Comment 3.6** — L161 & 203: Just mentioning "a selection of sites [...] to ensure adequate data quality" is a bit arbitrary

To clarify the site selection process, the methods & material section was revised:

"From the FLUXNET2015 dataset (Pastorello et al., 2020) and the ICOS '2018 drought initiative' dataset (Drought 2018 Team and ICOS Ecosystem Thematic Centre, 2019), sites were selected with adequate data quality (at least 1 year of carbon fluxes, dominated by observations with quality flag 1 or better), homogeneous land cover and limited disturbance due to management. This resulted in the 56 sites, listed in Table 2, and a total of 526 simulation years. 33 of these sites are dominated by forest land cover, whereas 18 are dominated by herbaceous vegetation and 5 are crop sites (the models are configured to run without management practices). The FLUXNET and ICOS data products had been pre-processed with the ONEFLUX processing pipeline (Pastorello et al., 2020). "

**Comment 3.7** — L165: Again of course not to be discussed here, could the non-specified management practices be an important explanation for the difference between diagnostic and prognostic model(s) in crop sites (e.g. Figure 3)?

Yes, some of the management practices are implicitly present in the forcing of the diagnostic model, whereas they are missing in the prognostic models. However, the performance of the prognostic models in crop sites is not significantly different to that in natural herbaceous sites. From these results, it is not clear to what extent missing management practices have an impact on the results. We add the following to the discussion:

"In the crop sites, management practices were missing in the prognostic models. In the mean annual cycle of LAI (Fig. 11), it is evident that the no harvest occurs. Despite this, the simulations of LE were not significantly less accurate compared to other land cover types. After harvest, LE consists largely out of bare soil evaporation. Though vegetation was still present in the models, the bulk LE was still reasonably accurate. More evident degradation of the results was found in GPP after harvest, which was overestimated. Even in the diagnostic model, where management practices were incorporated implicitly in the forcing variables, GPP was overestimated. Notably, despite the missing management practices in the prognostic models, the quality of the simulated LE and GPP (and their anomalies) was not significantly different from that in natural herbaceous sites. "

**Comment 3.8** — L168: Were the PFT and vegetation type info derived from the IGBP metadata of the flux network, or from remote sensing, or other sources?

The PFT was derived from the IGBP metadata of FLUXNET/ICOS. The text was revised:

" The test sites were classified per PFT (taken from the FLUXNET/ICOS IGBP metadata),... "

**Comment 3.9** — Table 2: Some sites (apparently especially crop sites, e.g. BE-Lon, DE-Kli and DE-RuS) are listed with very large LE corrections, that do not match what I thought I knew from past studies on their energy balance closure. I tried a detailed check on DE-RuS: The flux-weighted effective average factor between LE\_corr and LE is 1.44, somewhat lower (why) than the 1.47 in Table 2, but still far too high compared to any energy balance analysis carried out for this site in the past (e.g. 1.18 would result from Eder et al. 2015, DOI: 10.1175/JAMC-D-14-0140.1 which focused on summer months and 1.23 from Graf et al. 2020, doi.org/10.1098/rstb.2019.0524 with a study period matching the drought2018 dataset). Note that the current One-Flux product does to my knowledge not use details such as heat flux plate depth important to compute the energy balance closure, however even considering this the difference seems far too large, so it seems that LE\_corr from this dataset should be used with care.

The value of LE\_corr was calculated using aggregated daily LE values (LE\_CORR and LE\_F\_MDS). The reported value in the table is the mean (0.470523). If the median is used, we find 0.44 (0.437974). Perhaps that is the difference.

The calculation considers the full time period, and doesn't exclude the winter period, (during which LE is smaller and the relative correction is higher, see also Fig. S-1).

The uncertainty associated with the eddy covariance observations is discussed elaborately throughout this manuscript (e.g. section 4.1). We fully agree that the validation results need to be interpreted with care.

**Comment 3.10** — L182: Actually ICOS does have a standardized setup for soil moisture; however, the used dataset (drought2018) still mostly consists of so-called "legacy data" (i.e. voluntarily provided measurements with pre-ICOS set-ups). No need to mention it, just avoid the misleading wording.

We were not aware of this, good to know for the future! The sentence was changed as follows:

"Not all sites are equipped with soil moisture sensors, nor is there a standardized setup or post-processing for soil moisture in the datasets used for this study."

**Comment 3.11** — L196: From the way it is mentioned for LE and H and then a new paragraph starts, no EBC-based correction was assumed for NEE (and propagated to GPP and Reco)? Not that I would like to recommend it, just for clarity. Unfortunately even the correctability for LE and H is far from certain, but then depending on the assumed reasons it may or may not also apply to the CO2 flux (at least its turbulent part before WPL correction). It is nothing that can be done in a more certain way, but it is important to be aware of it later e.g. when LE and NEE show different model-observation biases. P.S.: It is nicely mentioned already in line 394, but may still leave the reader wondering here.

Indeed. For clarity we mention it here as well:

Figure S-1: Energy balance at DE-RuS

"Though some authors have recommended to correct the carbon fluxes in a similar way as the turbulent fluxes, such a procedure was not included in the processing pipeline (Massman and Lee, 2002; Gao et al., 2019, see also section 4). "

**Comment 3.12** — L221: Make clearer if the mean annual cycles are computed one per site across all its site-years (which implies that the deviations also include interannual variability, which is not a bad thing but one to be aware of)

The text was revised to clarify this, following the recommendation of the reviewer:

" The mean annual cycles were computed per site, across all its siteyears. "

**Comment 3.13** — L226-235 and Figures 5+6: Better explain for what the slope and for what the correlation was used. Comparing the text to the figure captions, I guess that the "Spearman slope" in the caption is wrong (slope yes but probably not between the rank-transformed variables, which is what "Spearman" would imply to me)?

There was indeed a mistake in the captions of the figures 5 and 6. "Spearman slope" should have been "slope". This was corrected.

The slope was used to evaluate the response of the surface fluxes to soil moisture and LAI. The analysis was done for the observations and the models. This allows to evaluate how well the sensitivity in the models resembles the observed sensitivity. It is explained in the manuscript as follows: "To assess the sensitivity of the fluxes to the state variables ( $S_e$  and LAI), the slope of the seasonal anomalies of the fluxes against the anomalies of the state variables was determined. This analysis was performed for the observations and the simulations, and compared. Note that the linear slope was used here, though a linear response is not necessarily expected (e.g. the response to soil moisture anomalies depend on a wet/dry regime). The goal of this analysis was to investigate whether LSM are capable of reproducing a similar relationship as found in the observations. Significant differences between the models were evaluated with the Wilcoxon signed-rank test."

Additional explanation was given in the result section, to help the reader:

" The sensitivity of the surface fluxes to soil moisture and LAI was quantified with a simple linear regression between their anomalies. The slope of these regressions indicates the strength of the response to the state variables. "

The error correlation was used to evaluate whether errors in the surface fluxes are associated with errors in the state variables (soil moisture and LAI).

" To evaluate whether errors in the state variables are associated with errors in the surface fluxes (or vice versa), the Spearman rank correlation between both was calculated."

Similarly, this was mentioned again in the result section:

" To evaluate the impact of the quality of  $S_e$  and LAI on the simulated surface fluxes, the Spearman correlation of the errors in the state variables and the fluxes was calculated."

**Comment 3.14** — L243: Maybe adding "independently" to the last sentence and mentioning it already at the start of subsection 2.3.2 would make it easier to understand.

This section was restructured to improve readability.

,,

**Comment 3.15** — L247: Here it is unclear whether the LE partitioning methods are just mentioned out of interest, or were applied in this study (which seems not to be the case according to the result section).

The transpiration derived from the observations was used to estimate the WUE in the sites. This was clarified:

" From the GPP and transpiration (Tr), the WUE was derived:

$$WUE = \frac{GPP}{Tr} \tag{S-1}$$

**Comment 3.16** — L255: Shouldn't this be visible in Fig. 2a? If accuracy corresponds with the bias (x axis) and precision with random errors (y axis), it would be more accurate to state that both models have the same accuracy but ISBA a slightly better precision. Sometimes accuracy is also used as a combined name corresponding to both,

systematic and random errors; then the statement is true but imprecise and the "significantly" seems a bit overstated (unless it refers to a successful statistical significance test of course).

Some confusion might arise from the use of the words "bias" and "accuracy". The terminology as it is used in this manuscript was defined at L250:

" The bias (ME) and accuracy (RMSE) of the simulated... "

To our knowledge it is not unusual to use bias and accuracy in this sense. Significantly refers indeed to a statistical significant difference with the Wilcoxon test. Note that this test is a pairwise comparison. These significant differences might not be evident from a figure like Fig. 2a. A scatter plot reveals this difference more clearly (see Fig. S-2).

Figure S-2: RMSE of the simulated LE with ISBA and ORCHIDEE.

**Comment 3.17** — Around L350, Figure 12: Are differences in the partitioning between drainage and runoff really interesting to discuss for models which were all run in uncoupled 1D mode? My (maybe wrong) expectation would be that it is a quite arbitrary function of model physics that only converges between models if horizontal neighbours with given slopes get a chance to communicate with each other.

The primary goal of showing the water balance is to provide further insight in the simulated water dynamics. Within the frame of this 1D experiment, the difference in water partitioning are deemed relevant, even though the models might behave differently in a coupled mode.

It is not clear why we could assume that the models would converge in a coupled/3D experiment.

**Comment 3.18** — L389: "caused by surface" looks a bit as if something was missed out here, maybe "... surface heterogeneities"?

Indeed, this was corrected.

Comment 3.19 — L396: Mentioning GPP and LE alongside each other with parenthesis does not fully capture the extend of the problem (see also comment on L196): Since LE was corrected for EBC non-closure (at least tried to, given the open questions correctly mentioned by the authors) while GPP was not, it could somewhat be expected that the mean difference model vs. obs is smaller (or more negative) for LE and larger (or zero or less negative) for GPP. This is exactly what we see in Fig. 2 (if the x axis is model - observation). Which might indicate (among other ways to explain more associated with model shortcomings of course) that the LE is overcorrected even by the current EBC correction. Note that to my knowledge (if I didn't overlook something) Gebler et al. (2015) do not report a better EC-lysimeter match by putting the whole deficit into LE, but by a correction conserving the evaporative fraction, which is similar to the Bowen ratio conserving correction by Pastorello et al. Others even suggest that most or all of the deficit might be related to sensible heat (Ingwersen et al. 2011, https://doi.org/10.1016/j.agrformet.2010.11.010), or found a good match with independent reference data without LE correction (e.g. Graf et al. 2014, https://doi.org/10.1002/2013WR( for the catchment water budget method). In general, a problem with the body of existing comparisons of eddy-covariance fluxes to independent reference methods is that the latter can have their own systematic errors (e.g. island effect in case of lysimeters, or different footprints of both systems) on a similar order of magnitude as the eddy-covariance energy balance closure gap, and that the (often quite definite) answers of the single studies are in conflict when comparing these studies with each other. Maybe (especially given the risk of a too large energy balance gap seen by the flux product as discussed in comment on table 2) it would even be interesting to see how the model-observation match without the energy balance correction is. Of course, the results would not completely reliably indicate an overcorrection / different source of the closure gap, but could also point to an unintended adaption of the models towards uncorrected eddy-covariance data during past validations.

We fully agree with this comment, highlighting the uncertainty related to the eddy covariance observations. The suggested additional references and discussion was incorporated in the text.

"Furthermore, some studies have indicated that the eddy covariance observations are closer to lysimeter data if the energy balance is closed by correcting LE only (Wohlfahrt et al., 2010). Considering this, the negative bias of the simulated LE (and GPP) in this study could be even underestimated. Conversely, others suggest that most or all of the deficit might be related to H (Ingwersen et al., 2011), or found a good match with independent reference data without LE correction (Graf et al., 2014). Validation results of the turbulent fluxes without energy balance closure correction are given in the supplement material. "

In addition, we provide some plots of the validation with and without EBC correction in the supplementary material (see Fig. S-3). Finally, the reference to Gebler et al. (2015) was indeed false. It should have been a reference to Wohlfahrt et al. (2010). This was corrected.

---

## Author Response (AR1)

**Rebuttal BG-2021-355**

Local scale evaluation of the simulated interactions between energy, water and vegetation in land surface model

Jan De Pue, José Miguel Barrios, Liyang Liu, Philippe Ciais, Alirio Arboleda, Rafiq Hamdi, Manuela Balzarolo, Fabienne Maignan, and Françoise Gellens-Meulenberghs Handling Associate Editor: Ivonne Trebs

July 15, 2022

**JC Calvet**

Many thanks for this interesting work. This is an excellent contribution to the continuous model intercomparison effort.

**Comment 1.1** — In order to allow the reproducibility of the simulations, it could be useful adding a Supplement with model namelists together with a Table listing key model parameters, their values, and how these values were determined.

For ISBA and ORCHIDEE, the namelists used for this experiment are added as supplement material. Additionally, tables listing some of the key plant physiology and soil physical parameter values are given as well. The architecture of the diagnostic model differs too much from the prognostic models to allow comparison of the plant physiological parameters. They are not included in these tables. The following text has been added to section 3.1:

" An overview of some key plant physiology parameters and soil physical parameters is given in the supplement material, along with full option namelists of the ISBA and ORCHIDEE runs to allow reproducibility.

Most of the vegetation parameters in ISBA are derived from the TRY plant traits database (Kattge et al., 2011; Delire et al., 2020). Parameters in ORCHIDEE are derived from the same database, but are regularly calibrated using various data types, including satellite observations, in situ fluxes and surface concentrations (e.g. Kuppel et al., 2012, 2014; MacBean et al., 2015; Peylin et al., 2016). Kuppel et al. (2014) used 78 FLUXNET sites to optimize parameters related to the NEE (net ecosystem exchange) and LE fluxes (see their Table S2). Hence, whereas the ORCHIDEE parameters were not optimized using the specific dataset of this study, a part of it may have been used formerly in this regard. Similarly, key parameters of the diagnostic model have been (indirectly) derived from the global network of eddy covariance stations (Garbulsky et al., 2010; Martínez et al., 2020). "

|                 | $V_{cmax}$ (mg | $m^{-2}s^{-1}$ ) | $SLA\ (m^2kgC^{-1})$ |      |
|-----------------|----------------|------------------|----------------------|------|
| PFT             | ISBA           | ORCH             | ISBA                 | ORCH |
| Temp DBF        | 2.60           | 1.14             | 15.4                 | 26.0 |
| Boreal ENF      | 2.80           | 1.02             | 5.0                  | 9.3  |
| Trop EBF        | 1.00           | 1.02             | 8.3                  | 15.3 |
| C3 crop         | 4.40           | 1.36             | 14.8                 | 26.0 |
| C4 crop         | 1.70           | 1.36             | 10.3                 | 26.0 |
| irr. Crop       | 3.40           | -                | 10.3                 | -    |
| C3 grass        | 3.40           | 1.14             | 14.0                 | 42.0 |
| C4 grass        | 1.70           | 1.14             | 5.7                  | 41.0 |
| Wetlands        | 3.40           | -                | 14.0                 | -    |
| Trop DBF        | 1.80           | 1.02             | 15.4                 | 26.0 |
| Temp EBF        | 2.60           | 0.91             | 8.3                  | 20.0 |
| Temp ENF        | 2.80           | 0.80             | 5.0                  | 9.3  |
| Boreal DBF      | 2.60           | 0.80             | 15.4                 | 26.0 |
| Boreal DNF      | 1.80           | 0.80             | 10.1                 | 19.0 |
| Boreal Grass    | 3.40           | -                | 14.0                 | -    |
| Deciduous Shrub | 2.40           | -                | 15.4                 | -    |

Table S-1: Key plant physiological parameters in ISBA and ORCHIDEE (for the conversion of  $V_{cmax}$  to  $g_m$  and  $A_{mmax}$ , see Delire et al. (2020). DBF = Deciduous Broadleaf Forest, ENF = Evergreen Needleleaf Forest, EBF = Evergreen Broadleaf Forest, DNF = Deciduous Needleleaf Forest.

| Texture class   | $\theta_s$    | $	heta_r$     | n    | $\alpha$               | $K_s$          | $\theta_{fc}$ |
|-----------------|---------------|---------------|------|------------------------|----------------|---------------|
|                 | $(m^3m^{-3})$ | $(m^3m^{-3})$ | (-)  | $(10^3 {\rm cm}^{-1})$ | $(cm\;d^{-1})$ | $(m^3m^{-3})$ |
| Sand            | 0.43          | 0.05          | 2.68 | 14.5                   | 7128.0         | 0.05          |
| Loamy Sand      | 0.41          | 0.60          | 2.28 | 12.4                   | 3501.6         | 0.07          |
| Sandy Loam      | 0.41          | 0.07          | 1.89 | 7.5                    | 1060.8         | 0.12          |
| Silt Loam       | 0.45          | 0.07          | 1.41 | 2.0                    | 108.0          | 0.24          |
| Silt            | 0.46          | 0.03          | 1.37 | 1.6                    | 60.0           | 0.26          |
| Loam            | 0.43          | 0.08          | 1.56 | 3.6                    | 249.6          | 0.17          |
| Sandy Clay Loam | 0.39          | 0.10          | 1.48 | 5.9                    | 314.4          | 0.17          |
| Silty Clay Loam | 0.43          | 0.09          | 1.23 | 1.0                    | 16.8           | 0.34          |
| Clay Loam       | 0.41          | 0.10          | 1.31 | 1.9                    | 62.4           | 0.27          |
| Sandy Clay      | 0.38          | 0.10          | 1.23 | 2.7                    | 28.8           | 0.27          |
| Silty Clay      | 0.36          | 0.07          | 1.09 | 0.5                    | 4.8            | 0.34          |
| Clay            | 0.38          | 0.07          | 1.09 | 0.8                    | 48.0           | 0.35          |

Table S-2: ORCHIDEE soil physical parameters for the Mualem-Van Genuchten model (derived from Carsel and Parrish (1988)),  $\theta_{fc}$  is the water content at field capacity.

| Texture class   | $\theta_s$    | $\Phi_s$ | b     | $K_s$          | $	heta_{fc}$  |
|-----------------|---------------|----------|-------|----------------|---------------|
|                 | $(m^3m^{-3})$ | (cm)     | (-)   | $(cm\;d^{-1})$ | $(m^3m^{-3})$ |
| Sand            | 0.40          | -12.10   | 4.05  | 1520.6         | 0.23          |
| Loamy Sand      | 0.41          | -9.00    | 4.38  | 1350.7         | 0.24          |
| Sandy Loam      | 0.44          | -21.80   | 4.90  | 299.5          | 0.32          |
| Silt Loam       | 0.49          | -78.60   | 5.30  | 62.2           | 0.46          |
| Silt            | 0.49          | -78.60   | 5.30  | 62.2           | 0.46          |
| Loam            | 0.45          | -47.80   | 5.39  | 60.0           | 0.39          |
| Sandy Clay Loam | 0.42          | -29.90   | 7.12  | 54.4           | 0.35          |
| Silty Clay Loam | 0.48          | -35.60   | 7.75  | 14.7           | 0.42          |
| Clay Loam       | 0.48          | -63.00   | 8.52  | 21.2           | 0.45          |
| Sandy Clay      | 0.43          | -15.30   | 10.40 | 18.7           | 0.36          |
| Silty Clay      | 0.49          | -49.00   | 10.40 | 8.9            | 0.46          |
| Clay            | 0.48          | -40.50   | 11.40 | 11.1           | 0.45          |

Table S-3: ISBA soil physical parameters for the Campbell model (derived from Clapp and Hornberger (1978)), $\theta_{fc}$  is the water content at field capacity.

| Texture class | $\theta_s$    | $	heta_{fc}$  | $	heta_{wilt}$ |
|---------------|---------------|---------------|----------------|
|               | $(m^3m^{-3})$ | $(m^3m^{-3})$ | $(m^3m^{-3})$  |
| Coarse        | 0.40          | 0.24          | 0.06           |
| Medium        | 0.44          | 0.35          | 0.15           |
| Medium fine   | 0.43          | 0.38          | 0.13           |
| Fine          | 0.52          | 0.45          | 0.28           |
| Very fine     | 0.61          | 0.54          | 0.34           |
| Organic       | 0.77          | 0.66          | 0.27           |
| Loamy         | 0.47          | 0.32          | 0.17           |

Table S-4: DiagMod soil physical parameters, based on the Wösten et al. (1999) PTF.

**Comment 1.2** — It should be clarified whether any of the in situ datasets used in this study as a reference was involved in previous model parameter tunings.

The global network of eddy covariance station is directly or indirectly (via derived products, e.g. FluxCom) involved in the parametrization of land surface models. This is the case in particular for ORCHIDEE and the diagnostic model. The manuscripts has been revised to clarify this (see previous response).

**Comment 1.3** — Finally, how were model simulations initialized (e.g. initial root-zone soil moisture conditions)?

The initialization of ISBA is described in section 2.1.2, L133:

" A spin-up period of 3 years was sufficient to eliminate effects from the initial model state on the surface fluxes (respiration is not analysed in this study). "

Concerning the initialization of ORCHIDEE, the following text was added to section 2.1.3:

" To initialize the simulations, a first spin-up phase was performed, where we cycled over the available FLUXNET years for at least 45 years. This enables to reach an equilibrium for the above-ground biomass and the water stocks and fluxes, as an initial state for the transient simulation. "

The diagnostic model does not require any initialization.

**Anonymous referee 1**

This work presents a systematic point-scale evaluation of two prognostic land surface models (LSMs) and one observation-driven diagnostic model across several measurement sites of the FLUXNET monitoring network. The assessment of models' performance is focused on the simulated latent heat flux (LE), gross primary production (GPP), soil moisture, and leaf area index (LAI). Overall, this manuscript aims at disentangling the relative role of soil moisture and leaf area index in explaining the key models' weaknesses in the simulation of the land-atmosphere water-energy-carbon exchanges.

This work addresses a subject of interest for the broad audience of BG and it has the potential to shed additional light and provide guidance to the LSM modelling community on the simulation of water-energy-carbon interactions and involved feedback mechanisms. Having said that, I think there are substantial improvements to be implemented in the manuscript before making any consideration for publication.

I provide a more detailed list of comments below:

**Comment 2.1** — The **introduction** of the manuscript is **too weak**, lacking the clear definition of the unresolved research questions that are behind this work. Those provided between lines 70-74 are, in my opinion, not scientifically relevant to justify the publication of this manuscript. In addition, **a mere models vs observations comparison is not per se a strong objective**; see lines 79-84. On the other hand, if the main objective of the manuscript is included in the last sentence of the introduction ("Given the degree of coupling in the current LSM, we try to disentangle the relation between key facets of the terrestrial vegetation in a holistic way"), authors should put **more emphasis on this aspect and less on the evaluation** of the models' performance.

As the reviewer points out, the manuscript balances between 2 aspects: model intercomparison and an analysis of the internal interactions. Rather than strictly reporting the model performances, we attempt to extend the analysis with some statistics to provide more insight in the origin of differences in the model performances. The soil moisture and LAI were identified as primary explanatory variables. With the given analysis, we find that some meaningful conclusions can be derived, though -as the reviewer points out- a more extensive analysis would allow more solid statements. We agree with this remark, it would be very relevant to test more model configurations. However, as pointed out in the next comments, there are some limitations to be considered.

Still, we agree with the reviewer and have followed this recommendation to shift the focus more towards the evaluation of the second (deemed more relevant) aspect. In this regard, the introduction was revised and the objectives are listed more clearly. We refer to the revised manuscript (where the revision is highlighted with tracked changes)

**Comment 2.2** — I found the relative role "assigned" to the diagnostic model in the intercomparison exercise not fully clear and justified. Specifically, If the scope of the work is to compare the coherence with respect a LSM prognostic approach (see lines 73-74), authors should have structured their comparison in a different way. That is,

they should have complemented the results of the **observation-driven** (i.e., remote sensing for LAI and ERA5 for soil moisture) diagnostic model with those obtained assuming the output of the two LSMs (i.e., soil moisture and LAI) as "observations". In this way authors should have been able to provide more stringent interpretations on the different models' performance and/or deficiency and eventually coherence.

This was a very valuable and constructive remark by the reviewer. We agree fully that more model configurations would make the conclusions of this study more solid. The 'functional evaluation' with the diagnostic model is a feasible way to achieve this (contrary to the simulations with imposed soil moisture and lai, see below). As illustrated in Fig. S-1, the Diagnostic model was used as a vehicle to test the prognostic LAI and soil moisture, resulting in a total of 7 runs per site (see Tab. S-5). The outputs for one site are shown in Fig. S-2. The results from this functional evaluation are well in line with the previous results. We refer to the manuscript for a more detailed discussion.

Additionally, prior to this analysis, the capacity of DiagMod to reproduce the results from the prognostic models (given the same forcings) was tested. The results are shown in the supplement material.

Figure S-1: Example of LAI and SWC timeseries (mean annual cycles, AU-DaP), used in the analysis with the diagnostic model.

Figure S-2: Resulting LE and GPP mean annual cycles with different model configurations.

|                | LAI      | SM       | LE    | GPP    |
|----------------|----------|----------|-------|--------|
| DiagMod        | CGLS     | ERA5     | 0.47  | 0.37   |
| laiISBA_swERA5 | ISBA     | ERA5     | 0.42  | 0.11*  |
| laiCGLS_swISBA | CGLS     | ISBA     | 0.45  | 0.27*  |
| laiISBA_swISBA | ISBA     | ISBA     | 0.45  | -0.08* |
| laiORCH_swERA5 | ORCHIDEE | ERA5     | 0.42  | -0.24* |
| laiCGLS_swORCH | CGLS     | ORCHIDEE | 0.24* | 0.29*  |
| laiORCH_swORCH | ORCHIDEE | ORCHIDEE | 0.27* | -0.50* |

Table S-5: Median Nash-Sutcliffe model efficiency index of the DiagMod runs (to evaluate the prognostic LAI and soil moisture). Results presented for all sites, significant differences (Wilcoxon p